# Hydrophilic nanoparticles that kill bacteria while sparing mammalian cells reveal the antibiotic role of nanostructures

Yunjiang Jiang[1,2], Wan Zheng[1], Keith Tran[1], Elizabeth Kamilar[1], Jitender Bariwal [1], Hairong Ma[1] & Hongjun Liang [1✉]

To dissect the antibiotic role of nanostructures from chemical moieties belligerent to both bacterial and mammalian cells, here we show the antimicrobial activity and cytotoxicity of nanoparticle-pinched polymer brushes (NPPBs) consisting of chemically inert silica nanospheres of systematically varied diameters covalently grafted with hydrophilic polymer brushes that are non-toxic and non-bactericidal. Assembly of the hydrophilic polymers into nanostructured NPPBs doesn't alter their amicability with mammalian cells, but it incurs a transformation of their antimicrobial potential against bacteria, including clinical multidrug-resistant strains, that depends critically on the nanoparticle sizes. The acquired antimicrobial potency intensifies with small nanoparticles but subsides quickly with large ones. We identify a threshold size ($d_{silica}$ ~ 50 nm) only beneath which NPPBs remodel bacteria-mimicking membrane into 2D columnar phase, the epitome of membrane pore formation. This study illuminates nanoengineering as a viable approach to develop nanoantibiotics that kill bacteria upon contact yet remain nontoxic when engulfed by mammalian cells.

[1] Department of Cell Physiology & Molecular Biophysics, Center for Membrane Protein Research, School of Medicine, Texas Tech University Health Sciences Center, Lubbock, TX 79430, USA. [2] Present address: BayRay Innovation Center, Shenzhen Bay Lab, Shenzhen, Guangdong Province 518107, China. ✉email: H.liang@ttuhsc.edu

Antibacterial nanomaterials, or nanoantibiotics, are emerging contenders to fend off drug-resistant bacteria when conventional antibiotics fail[1,2]. This new arena of antibiotics encompasses a plethora of nanostructured pathogen fighters, such as nanocarriers loaded with antibiotics[3], metal[4,5] or metal oxide nanoparticles[6,7], organic–inorganic composite nanoparticles[8–10], graphene or graphene oxide[11–13], carbon nanotubes[14], dendrimers[15], self-assembled micelles[16], unimolecular micelles[17], supramolecular nanostructures[18,19], and polymer molecular brushes (i.e., bottlebrush polymers)[20–23]. Notwithstanding the individual differences attributed to the antimicrobial behaviors among the different nanoantibiotics, they share the similarity of having at least one physically confined dimension in the nanoscale. There are direct and indirect evidences that suggest size-dependent antimicrobial activities exist at the nanoscale[4–8,11–14], and nanostructures themselves are widely and sometimes blithely speculated to instigate added benefits in killing bacteria. However, many reported antimicrobial activities of nanoantibiotics resonate with the more general size-dependent toxicity as shown by the generation of reactive oxygen species (ROS), release of heavy metal ions, or increase of the specific hydrophobic surface area etc. that all become prevalent at the nanoscale, which help kill bacteria but also debilitate mammalian cells. Additionally, while patterned nanostructure arrays on engineered surfaces that act collectively to deter bacteria adhesion or kill bacteria through physico-forces or mechano-forces are known[24,25], whether individual inert nanostructures play any role on defining the encounter between nanoantibiotics and bacteria that seals the dour fate of the microbes is unclear. The lack of insight into how nanomaterials encroach bacteria casts a shadow over the prospect of nanoantibiotics, as the same mode of action that dooms bacteria may also give rise to undesirable cytotoxicity. Although the antibacterial potency of nanoantibiotics needs not translate to nanotoxicity against mammalian cells, the perceived boundary between the two is blurred nonetheless because on one hand, the inauspicious cytotoxicity and environmental hazard associated with many candidates of nanoantibiotics ranging from metal or metal oxide nanoparticles[26,27] to graphene, graphene oxide[28], and carbon nanotubes[29] have been well documented; on the other hand, the chemical moieties bestowed upon many nanoantibiotics with the aim to kill bacteria are also pernicious to mammalian cells. For instance, the hydrophobicity believed to be a critical antibacterial trait for membrane-active antimicrobials[30–32], which include most antibacterial dendrimers[15], micelles[16,17], supramolecular nanostructures[18,19], bottlebrush polymers[20,21], and organic–inorganic composite nanoparticles[8–10] under development, would also impartially damage mammalian cells because of the similar mode of membrane disruptions induced by hydrophobic interactions. Numerous chemical variations have been tested in search of a delicate yet unquantified hydrophobic-cationic balance in hopes of selectively killing bacteria without damaging mammalian cells[30–32], but many of the efforts were to little avail when it comes to clinically viable candidates.

In order for nanoantibiotics to stay relevant in the clinical battlegrounds fighting bacterial infections, it is imperative to dissect the antibacterial role of benign nanostructures from belligerent chemical moieties that indiscriminately target both bacteria and mammalian cells. Unlike mammalian cells that enlist various endocytosis pathways to engulf nanoparticles coming in contact[33], bacteria in general (although exception may exist[34]) are not known to uptake nanomaterials. Although disruption of bacterial membrane was observed with many nanoantibiotics[9–12,14–23], it remains to be clarified whether it is the nanostructure or the toxic chemical moiety associated with the nanoantibiotics that helps deliver this lethal blow. If benign nanostructures can be tailored to help bust bacterial membranes while sparing mammalian cells, we shall envisage a nanoengineering approach in which rationally designed nanoantibiotics carrying biocompatible chemical moieties kill bacteria upon contact yet remain nontoxic when engulfed by mammalian cells.

To illustrate the feasibility of this concept, we study here the antimicrobial activities and cytotoxicity of a series nanoparticle-pinched polymer brushes (NPPBs) consisting of chemically inert silica nanospheres of systematically varied diameters ($d_{silica}$ ~ 7–270 nm) covalently grafted with hydrophilic linear-chain brush polymers that by themselves are non-toxic and non-bactericidal. We demonstrate that nanostructures themselves have the potential to radically alter how hydrophilic linear-chain brush polymers interact with bacteria, which transforms the NPPBs into potent antibiotics without changing their benevolent stance on mammalian cells. We further identify a threshold size of silica nanospheres (i.e., $d_{silica}$ ~ 50 nm) beneath which the acquired antimicrobial activity of NPPBs significantly intensifies. Coincidentally, synchrotron small angle x-ray scattering (SAXS) studies reveal that only beneath this threshold, NPPBs remodel the microbial-mimicking membrane through a topological transition to form a 2D columnar phase, the epitome of membrane pore formation. In contrast to their antimicrobial activities, the hydrophilic NPPBs show low hemolysis to human red blood cells (HRBCs) and low cytotoxicity to human embryonic kidney 293 (HEK-293) cells regardless of their sizes. Through a diverse array of microstructure characterization and biological assays, we unveil the critical roles of both the intrinsic curvature of cell membrane lipids and nanoparticle size that synergistically define the initial encounter of hydrophilic nanoparticles with bacterial and mammalian cells, respectively, which sets apart their different paths toward different cellular fates. Taken together, this work illuminates nanoengineering as a viable approach to develop nanoantibiotics that selectively kill bacteria upon contact yet remain nontoxic to mammalian cells.

## Results and discussion

**Synthesis and characterization of hydrophilic NPPBs with well-defined sizes.** To study the antibiotic role of nanostructures, we designed model NPPBs consisting of inert silica nanospheres of systematically varied diameters grafted with poly(4-vinyl-N-methylpyridine iodide) (P4MVP) brushes of similar degree of polymerization (i.e., DP ~ 32). Both silica and P4MVP have been explored for biomedical applications. For example, P4MVP has been studied for gene delivery[35,36]; silica is listed amongst the "generally regarded as safe" substances by the Food and Drug Administration (ID Code: 14808-60-7)[37], and the long-term effect of silica nanoparticles on human health is still under scrutinization[38]. The metabolization and ultimate fate of NPPBs in human body are not known, but the underlying mechanism of nanostructure-induced transformation of antimicrobial activity revealed using this model system should still have broad implications.

To synthesize the model NPPBs, silica nanospheres of well-controlled sizes carrying surface derivatives of the initiator moiety (i.e., α-bromoisobutyryl bromide) for atom transfer radical polymerization (ATRP) were first synthesized. Surface-initiated ATRP (SI-ATRP) was then performed to graft well-defined poly(4-vinylpyridine) (P4VP) brushes on the silica nanospheres (i.e., SiO₂@P4VP), followed by a quaternization reaction with methyl iodide[39] to turn P4VP into hydrophilic and cationic P4MVP brushes (Fig. 1a). Experimental details on the synthesis and characterization of model NPPBs are included in Supplementary Information.

We used a series of characterization methods such as transmission electron microscopy (TEM), gel permeation

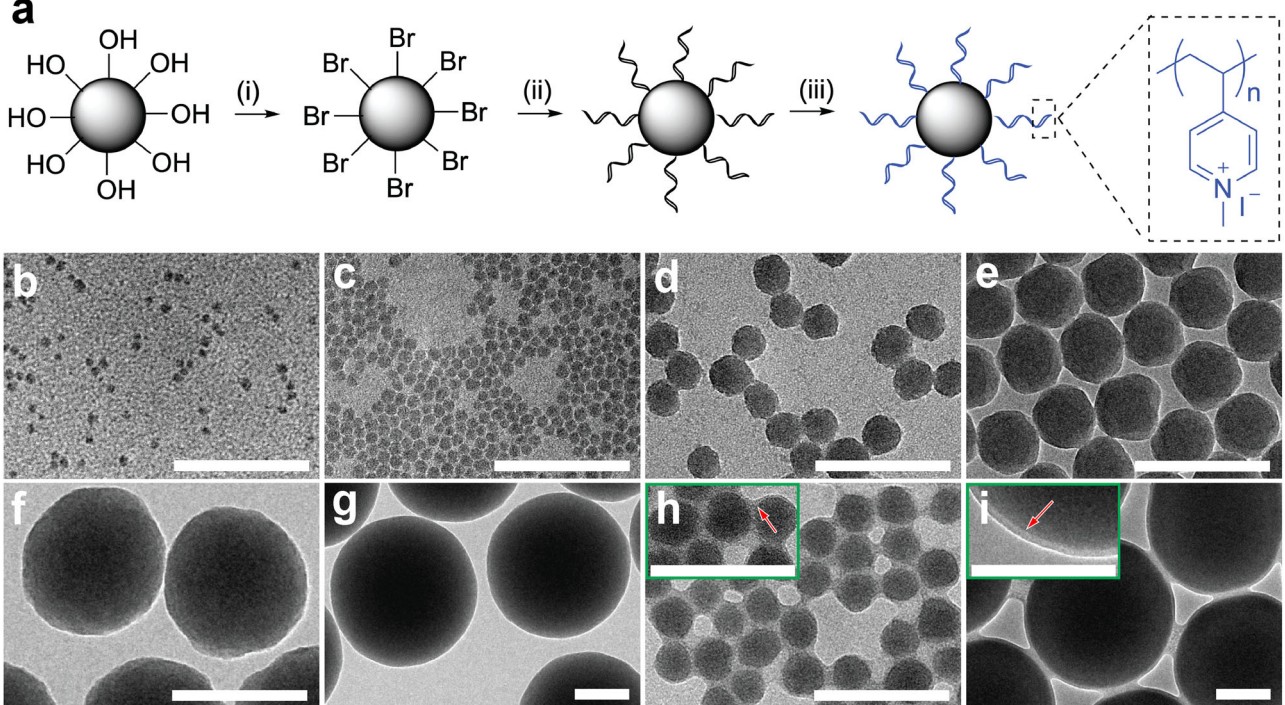

**Fig. 1 Synthesis and characterization of model NPPBs. a** Schematic of the synthesis design. (i) Silica nanospheres of well-controlled sizes carrying surface derivatives of ATRP initiators were first prepared, followed by (ii) the growth of well-defined P4VP brushes on the silica nanospheres via SI-ATRP, and (iii) the formation of NPPBs by converting the P4VP brushes into hydrophilic and cationic P4MVP via a quaternization reaction. **b–i** Representative TEM pictures (scale bar: 100 nm) of the silica nanospheres of systematically increasing diameters, i.e., **b** S7 ($d = 7.2 \pm 1.0$ nm), **c** S10 ($d = 9.9 \pm 1.1$ nm), **d** S25 ($d = 25.8 \pm 2.4$ nm), **e** S50 ($d = 45.6 \pm 2.0$ nm), **f** S110 ($d = 112 \pm 10$ nm), and **g** S270 ($d = 270 \pm 16$ nm), respectively, are shown and compared to those grafted with the P4VP brushes, i.e., **h** S25@P4VP$_{31}$ and **i** S270@P4VP$_{32}$. Note the P4VP brushes appear as a light-gray shell (pointed by red arrow in the insets) on the dark silica nanospheres. Similarly sized silica nanospheres as shown in **b–i** are consistent across all TEM pictures of individual samples taken in different experiments ($n = 5$).

**Table 1 The structural characteristics of model NPPBs.**

| NPPBs | $d_{silica}$ (nm)[a] | DP[b] | DP[c] | PDI[d] | Graft density (chain/nm²)[e] |
|---|---|---|---|---|---|
| S7–P35 | $7.2 \pm 1.0$ | 35 | 33 | 1.03 | 0.29 |
| S10–P33 | $9.9 \pm 1.1$ | 33 | 35 | 1.19 | 0.38 |
| S25–P31 | $25.8 \pm 2.4$ | 31 | 35 | 1.21 | 0.57 |
| S50–P29 | $45.6 \pm 2.0$ | 29 | 33 | 1.29 | 0.62 |
| S110–P34 | $112 \pm 10$ | 34 | 33 | 1.12 | 0.75 |
| S270–P32 | $270 \pm 16$ | 32 | 34 | 1.10 | 0.90 |

[a]Determined by TEM analysis (standard deviations are shown; $n = 20$ for all but S270–P32 ($n = 17$)).
[b]Determined by NMR.
[c]Determined by conversion analysis.
[d]Determined by GPC.
[e]Determined by TGA.

chromatography (GPC), nuclear magnetic resonance (NMR) spectroscopy, Fourier transform infrared (FTIR) spectroscopy, and thermal gravimetric analysis (TGA) together with brush cleavage experiments to determine the silica nanosphere diameter ($d_{silica}$), polymer brush size (DP) and polydispersity index (PDI), the degree of quaternization, and graft density (Supplementary Figs. 5–7). We confirmed the successful synthesis of six model NPPBs consisting of P4MVP brushes of a similar size (i.e., DP = $32 \pm 3$) covalently grafted onto silica nanospheres of systematically varied diameters (i.e., $d_{silica} \sim$ 7–270 nm). Representative TEM pictures of the silica nanospheres before (Fig. 1b–g) and after SI-ATRP of the P4VP brushes (Fig. 1h, i) are

shown. The P4VP brush layer appears as a light-gray shell (pointed by red arrow) overlaid on top of the dark silica nanospheres due to their electron density contrast. The thickness of this shell expands when the P4VP brush size increases[39]. All NPPBs are positively charged with similar zeta potentials (Supplementary Fig. 8), which also reflect their similarly sized P4MVP brushes. As we reported before[39], the zeta potential of NPPBs (i.e., the charge state of P4MVP brushes) is independent on pH.

A summary of the structural characteristics of all model NPPBs is listed in Table 1, in which the hydrophilic nanoparticles were individually named as "Sm–Pn", where "S" and "P" stand for the silica nanosphere and polymer brush, respectively, with $m$ and $n$ denoting their sizes in nanometer or DP. For example, S25 refers to bare silica nanospheres of $d_{silica} = 25$ nm, and S25–P31 refers to the hydrophilic NPPB consisting of S25 covalently grafted with P4MVP brushes of DP = 31. We used the DPs of free P4VP grown simultaneously in the same synthesis batches of SiO$_2$@P4VP to represent the brush sizes on individual NPPBs, as little difference exists between the surface-bound brushes versus free polymers grown simultaneously during the synthesis of polymer brushes via controlled/"living" polymerization[39]. We further validated their similarity by cleaving the P4VP brushes and comparing their GPC profiles with those of the free P4VP polymers grown concurrently in the same synthesis batch (Supplementary Fig. 6). Considering that the P4MVP brushes are polyelectrolytes, we estimated their radius of gyration (i.e., for DP = 32) to be ~2.3 nm and critical graft density for mushroom-to-brush transition to be ~0.06 chain/nm², which is far below that of all model NPPBs (Table 1).

**Antimicrobial activity and cytotoxicity of model NPPBs.** We used standard bacteria killing and inhibition assays[22,40,41] against two representative strains from each bacterial family, Gram− *Escherichia coli* (i.e., *E. coli*) and *PA14*, and Gram+ *Staphylococcus aureus* (i.e., *S. aureus*) and *MU50*, respectively, in which *PA14* (the tobramycin and gentamycin-resistant *Pseudomonas aeruginosa*) and *MU50* (the methicillin, oxacillin, and vancomycin-resistant *S. aureus*) are clinical multidrug resistant bacterial strains, to obtain the minimum bactericidal concentration (MBC) and inhibitory concentration (MIC). We quantified cytotoxicity by standard hemolysis and MTT assays against HRBCs and HEK-293 cells, respectively, to obtain $HC_{50}$ (i.e., the concentration at which 50% of the HRBCs are lysed)[22] and $IC_{50}$ (i.e., the concentration at which the viability of HEK-293 cells is reduced by 50%)[23]. We also used live/dead cell staining assays to directly observe the wellbeing of bacteria and HEK-293 cells[22,23].

As controls for NPPBs, the hydrophilic brush polymer $P4MVP_{28}$ (i.e., P28) by itself is non-hemolytic and does not show MIC against Gram− *E. coli* and *PA14*, nor MBC against *E. coli* up to 512 μg/mL that we tested[22]. Although it reaches MIC against Gram+ *S. aureus* (MIC = 24 μg/mL) and *MU50* (MIC = 128 μg/mL), which is consistent with previous reports that cationic compounds are in general strongly bacteriostatic against Gram+ bacteria[42,43], no MBC is observed against *S. aureus* up to 512 μg/mL that we tested[22]. The bare silica nanospheres also show no MIC nor MBC against both Gram− *E. coli* and Gram+ *S. aureus* up to 4000 μg/mL that we tested (Supplementary Fig. 9). Besides their cordiality with bacteria, both P28 and silica nanospheres by themselves show no cytotoxicity when tested by hemolysis[22] and live/dead cell staining assays (Supplementary Fig. 9), respectively.

Formation of NPPBs by grafting the non-toxic and non-bactericidal hydrophilic brush polymers onto silica nanospheres doesn't alter their amicability with mammalian cells, but it incurs a collective transformation of their antimicrobial potential against both Gram+ and Gram− bacteria, including clinical multidrug-resistant *PA14* and *MU50* strains (Fig. 2a, b). The hemolysis (Fig. 2c) and MTT assays (Fig. 2f) show that the resultant hydrophilic NPPBs behave similarly as the P28 brush by itself or the silica nanospheres alone, revealing no $HC_{50}$ against HRBCs nor $IC_{50}$ against HEK-293 cells. In contrast, a nanostructure-induced transformation of antimicrobial activity shows up in both the MBC (Supplementary Fig. 10) and MIC assays against Gram− (Fig. 2a) and Gram+ bacteria (Fig. 2b), respectively. A list of MICs, MBCs, $HC_{50}$, and $IC_{50}$ values of model NPPBs are summarized in Table 2. Interestingly, the acquired antimicrobial potential due to the assembly of polymer brushes into nanostructures depends critically on the nanoparticle size: it intensifies with smaller nanoparticles ($d_{silica} \leq 50$ nm) but subsides quickly with larger ones ($d > 50$ nm). For example, the MICs for S7–P35 and S50–P29 against *E. coli* are 32 and 64 μg/mL, respectively, but the MICs for S110–P34 and S270–P32 against *E. coli* are increased dramatically to 512 and 1000 μg/mL, respectively (Table 2). Given that the graft density of P4MVP brushes differs in NPPBs of different sizes (Table 1), the antimicrobial activities of NPPBs are further normalized to P4MVP brush concentrations for individual NPPBs with different specific surface areas, which too reveal a critical nanoparticle-size dependency, as examples of the normalized MICs shown for *E. coli* (Fig. 2d) and *S. aureus* (Fig. 2e), respectively.

This nanoparticle-size dependent antimicrobial potential is also evident in MBC assays (Supplementary Fig. 10 and Table 2) and in plain sight under confocal microscope: when the live and dead bacteria are stained in green and red[22], respectively, the wellbeing of *E. coli* (Fig. 2g–j) and *S. aureus* (Fig. 2k–n) incubated with S10–P33 (Fig. 2g, k), S25–P31 (Fig. 2h, l), S110–P34 (Fig. 2i, m), and S270–P32 (Fig. 2j, n) at the same dose of NPPBs (i.e., 32 μg/mL) is visually stunning. At $d_{silica} \leq 50$ nm, all *E. coli* (Fig. 2g, h) and *S. aureus* (Fig. 2k, l) are killed, whereas most *E. coli* (Fig. 2i, k) and *S. aureus* (Fig. 2m, n) are alive when NPPBs with $d_{silica} > 50$ nm are used. A threshold nanoparticle size ($d_{silica} \sim 50$ nm) appears to exist and roughly delineates the boundary between NPPBs of weak and strong antimicrobial potential. The presence of this boundary is not surprising for Gram+ bacteria because they are encapsulated by a thick nanoporous peptidoglycan layer that has a "mesh size" between ca. 5 to 50 nm[44,45]. As we discovered before, this nanoporous peptidoglycan capsule acts as a selective filter that only allows small nanoantibiotics ($d \leq 50$ nm) to cross and take actions but precludes large nanoantibiotics ($d > 50$ nm) from gaining access to the bacterial membrane[22,23]. What remains puzzling is the existence of this boundary for Gram− bacteria as their thin peptidoglycan layer is sandwiched between the bacterial outer and inner membranes. Considering that the hydrophilic NPPB nanoparticles ($d_{silica} \sim 7$–270 nm) would be too large to pass through bacterial membrane channels such as porins or LamB[46], they would have to disrupt the outer membrane of Gram− bacteria, which would kill the bacteria in the first place, before gaining access to their peptidoglycan layer. The observed nanoparticle size-dependent antimicrobial activities of the hydrophilic NPPBs against Gram− bacteria thus underscores a different yet unidentified mechanism at play.

**Hydrophilic NPPBs are membrane-active antimicrobials that target bacteria with size-dependent membrane disruption activity while sparring mammalian cells.** To gain more mechanistic insight on the different modes of actions that define the encounter between NPPBs and bacteria or mammalian cells, we used model giant unilamellar vesicles (GUVs) to mimic bacteria and mammalian cells, respectively. Given the dynamic membrane composition, membrane asymmetry, and cell wall differences among mammalian and bacterial cells, it is challenging to rely on simple GUVs to inform all aspects of responses ensued from mammalian or bacterial cells interacting with exogenous substances[47,48]. We chose GUVs to shed light on the initial membrane remodeling event due primarily to their membrane lipid difference when bacteria or mammalian cells encounter NPPBs, and cross-checked the utility of this approach by comparing the data obtained from model GUVs with those from live cells. An important difference between mammalian and microbial membranes lies in their lipid compositions[32,49–52]. Unlike mammalian membranes that consist predominantly of lipids with zero intrinsic curvature (e.g., phosphatidylcholine (PC) lipids), microbial membranes are laden with lipids of negative intrinsic curvature (e.g., phosphatidylethanolamine (PE) lipids)[51,52]. Lipid bilayers enriched with zwitterionic PE and PC lipids, respectively, have been widely used as valuable models to mimic bacterial and mammalian membranes[53,54]. Following this tradition and the pioneer works by Wong and colleagues[55,56], we chose model GUVs comprised of binary mixtures of anionic 1,2-dioleoyl-*sn*-glycero-3-phospho-(1′-*rac*-glycerol) (DOPG) and zwitterionic 1,2-dioleoyl-*sn*-glycero-3-phosphocholine (DOPC) or 1,2-dioleoyl-*sn*-glycero-3-phosphoethanolamine (DOPE). Both DOPG and DOPC have zero intrinsic curvature, whereas DOPE has a negative intrinsic curvature. We used 20/80 (molar ratio) DOPG/DOPE and DOPG/DOPC, respectively, to mimic the PE-rich microbial and PC-rich mammalian membranes by keeping the membrane charge density the same, hence eliminating any other variable in the model system.

To illustrate whether NPPBs encroach bacterial or mammalian cell membranes, we prepared bacteria-mimicking and

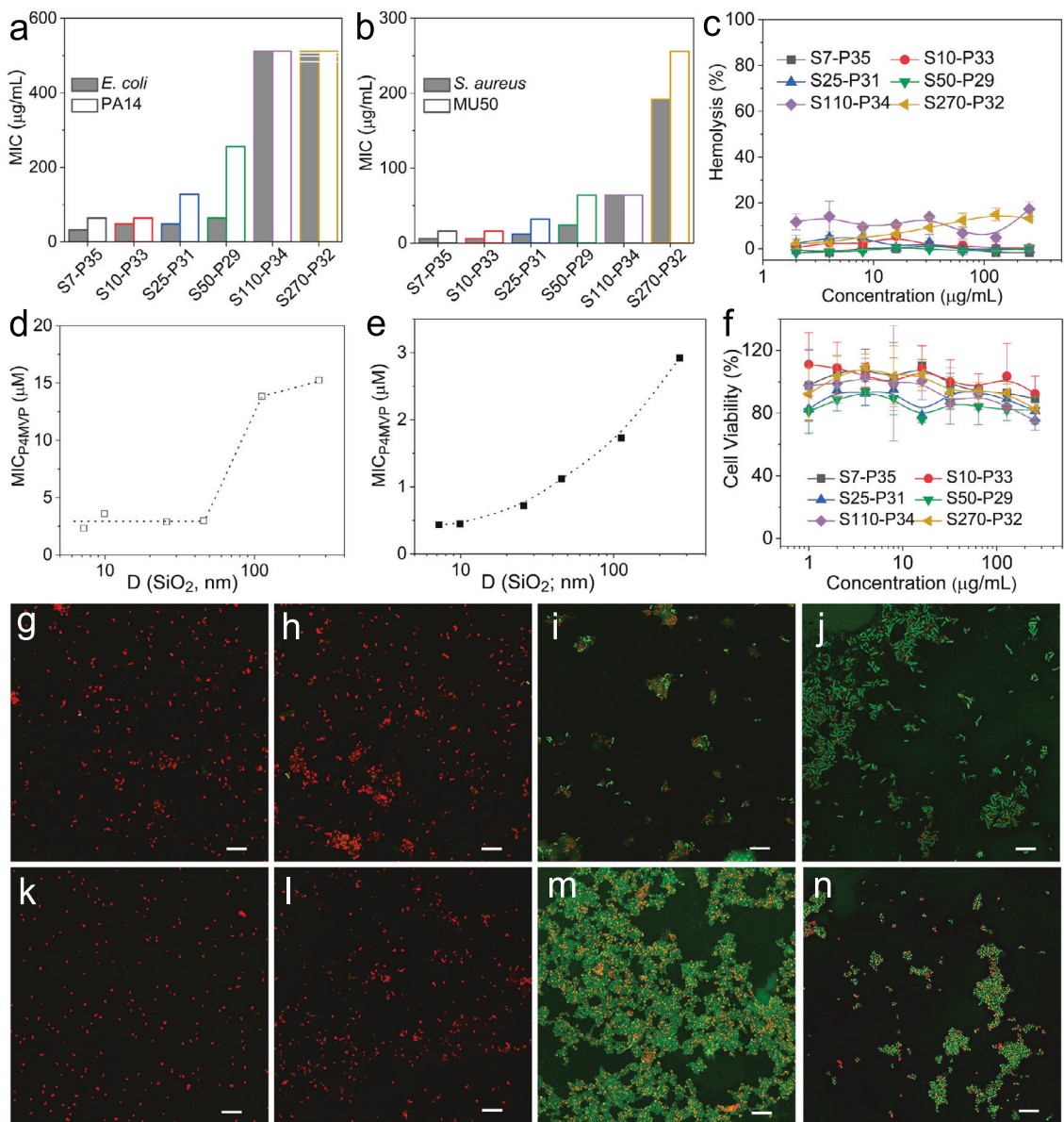

**Fig. 2 The biological activities of model NPPBs.** Although the hydrophilic NPPBs show no $HC_{50}$ against HRBCs (**c**) nor $IC_{50}$ against HEK-293 cells (**f**) (Error bars = Standard Deviation ($n = 8$) in both **c** and **f**), they show nanoparticle-size dependent bacteriostatic activities against **a** Gram− *E. coli* and *PA14*, and **b** Gram+ *S. aureus* and *MU50*. This nanostructure-induced transformation of antimicrobial activity intensifies with smaller NPPBs ($d_{silica} \leq 50$ nm) but subsides quickly with larger ones ($d_{silica} > 50$ nm). The same nanoparticle-size dependency is further revealed when the MICs are normalized to P4MVP brush concentration for NPPBs of different sizes and specific surface areas as shown in **d** Gram− *E. coli*, and **e** Gram+ *S. aureus*, respectively. The bactericidal activities of model NPPBs follow a similar nanoparticle-size dependency (Supplementary Fig. 10), which is on clear display under confocal microscope by live/dead assays of *E. coli* (**g**–**j**) and *S. aureus* (**k**–**n**) incubated with S10–P33 (**g**, **k**), S25–P31 (**h**, **l**), S110–P34 (**i**, **m**), and S270–P32 (**j**, **n**) at the same NPPB concentration of 32 μg/mL (Scale bar: 10 μm). Similar snapshots demonstrating the nanoparticle-size dependent bactericidal activities as shown in **g**–**n** are consistent across all confocal microscopy pictures of individual samples taken in different experiments ($n = 5$).

**Table 2 Antimicrobial activity and toxicity of model NPPBs (unit: μg/mL).**

| NPPBs | MIC | | | | MBC | | Cytotoxicity | |
|---|---|---|---|---|---|---|---|---|
| | *E. coli* | *PA14* | *S. aureus* | *MU50* | *E. coli* | *S. aureus* | $HC_{50}$ | $IC_{50}$ |
| S7–P35 | 32 | 64 | 6 | 16 | 48 | 16 | >512 | >512 |
| S10–P33 | 48 | 64 | 6 | 16 | 16 | 8 | >512 | >512 |
| S25–P31 | 48 | 128 | 12 | 32 | 32 | 16 | >512 | >512 |
| S50–P29 | 64 | 256 | 24 | 64 | 32 | 16 | >512 | >512 |
| S110–P34 | 512 | 512 | 64 | 64 | 128 | 64 | >512 | >512 |
| S270–P32 | 1000 | 1500 | 192 | 256 | >128 | 128 | >512 | >512 |

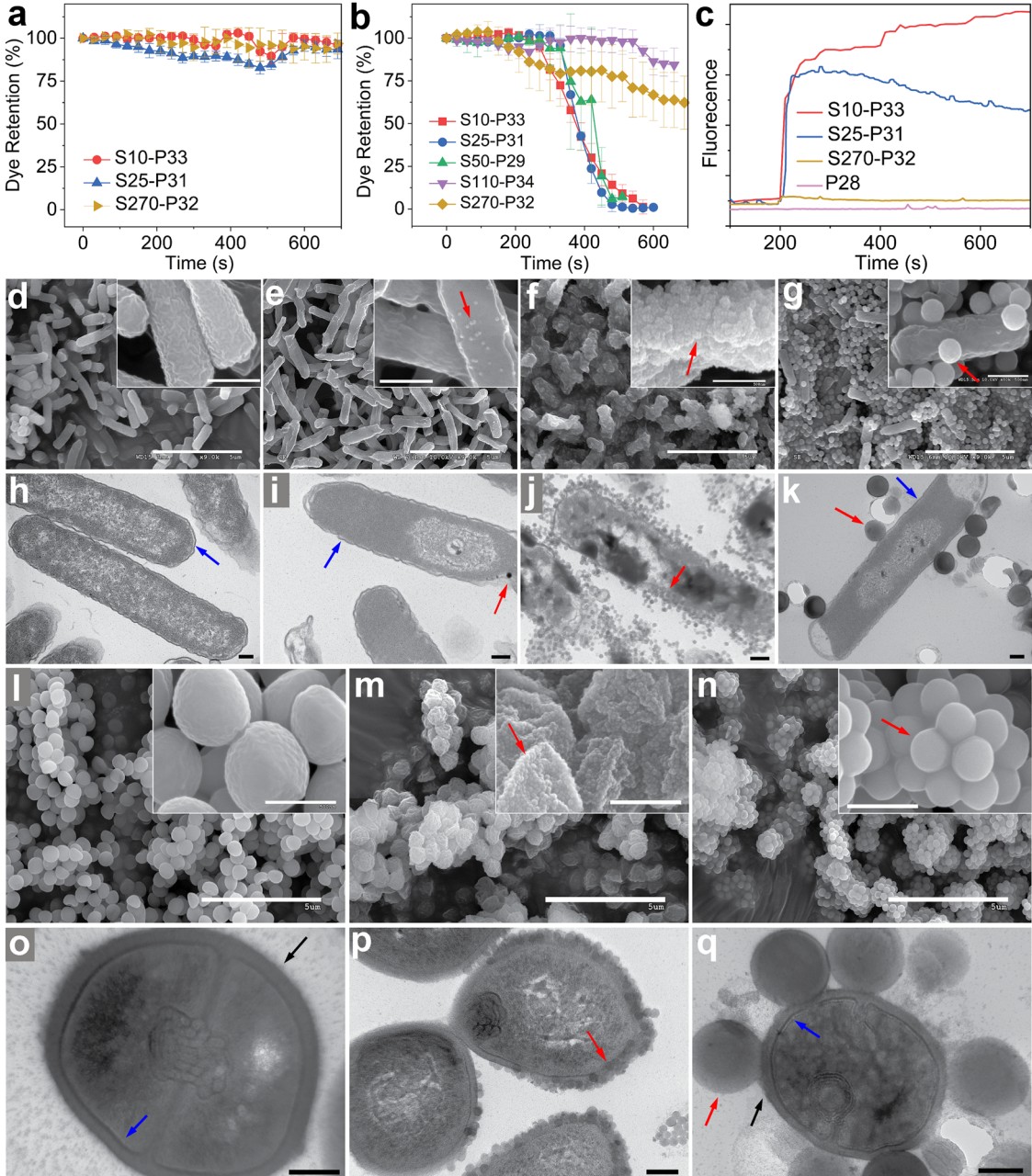

**Fig. 3 Hydrophilic NPPBs are membrane-active antimicrobials that selectively disrupt bacterial instead of mammalian membranes with size-dependent activities. a**, **b** Fluorescein release from GUVs that mimic mammalian cell and bacteria, respectively, incubated with model NPPBs (Error bars = Standard Deviation ($n = 3$) in both **a** and **b**). **c** Membrane permeation assays of *E. coli* incubated with model NPPBs and P28 control. **d**–**k** SEM (**d**–**g**) and cross-sectional TEM (**h**–**k**) of *E. coli* control (**d**, **h**), and *E. coli* incubated with S25 (**e**, **i**), S25–P31 (**f**, **j**), and S270–P32 (**g**, **k**), respectively. **l**–**q** SEM (**l**–**n**) and cross-sectional TEM (**o**–**q**) of *S. aureus* control (**l**, **o**), and *S. aureus* incubated with S25–P31 (**m**, **p**), and S270–P32 (**n**, **q**), respectively. The membrane, peptidoglycan layer, and nanoparticles are indicated by blue, black, and red arrows, respectively. Scale bars: 5 μm (SEM), 500 nm (SEM inset), and 200 nm (TEM). Similar SEM and TEM micrographs as shown in **d**–**q** are consistent across all pictures of individual samples taken in different experiments ($n = 6$).

mammalian cell-mimicking GUVs loaded with fluorescein and used confocal microscopy to examine membrane integrity by monitoring the dye leakage from GUVs exposed to individual NPPBs. Our previous study has shown that the P28 control doesn't cause dye leakage from both types of GUVs[22]. Examples of time-lapse confocal microscopy images of GUVs interacting with model NPPBs are shown in Supplementary Fig. 11. No dye leakage is observed from mammalian cell-mimicking GUVs incubated with NPPBs of any size (Fig. 3a). This is consistent with hemolysis assays that revealed very weak hemolytic activity for all

NPPBs (Fig. 2c), suggesting that the hydrophilic nanoparticles don't rupture mammalian membranes. The compatibility of NPPBs to mammalian cells is further supported by the MTT assay (Fig. 2f): even though endocytosis may occur when HEK-293 cells encounter the hydrophilic nanoparticles, the viability of the cells still suggests that their membrane integrity is not compromised because loss of homeostasis would lead to cell death. In contrast, when bacteria-mimicking GUVs interact with the NPPBs, rapid and complete dye leakage indicative of membrane pore formation occurs when smaller NPPBs ($d_{silica} \leq$

50 nm) are used, whereas small and slow dye release revealing mild deterioration of membrane integrity over time is observed for larger NPPBs ($d_{silica} > 50$ nm) (Fig. 3b). Taken together, the dye leakage assays suggest that NPPBs remodel cell membranes with different modes of interaction depending on both the intrinsic curvatures of membrane lipids and the hydrophilic nanoparticles sizes.

The relevance of probing cell membrane integrity using model GUVs and dye leakage assays is further vindicated by the bacterial membrane permeation assay (Fig. 3c), MIC and MBC assays (Fig. 2 and Table 2), as well as the scanning electron microscopy (SEM) and cross-sectional TEM studies of both Gram− *E. coli* (Fig. 3d–k) and Gram+ *S. aureus* (Fig. 3l–q). Permeation of hydrophobic fluorescent probe 1-N-phenylnaphthylamine into disrupted outer membrane of Gram− bacteria causes a prominent increase of its fluorescent emission, which serves as a telltale sign to indicate whether the membrane disruption occurs[57]. The membrane permeation assay (Fig. 3c) confirms that while *E. coli* membrane does not break in the presence of P28 control or larger NPPBs such as the S270–P32, membrane disruption does occur when *E. coli* cells encounter smaller NPPBs (i.e., $d_{silica} \leq 50$ nm), as indicated by a dramatic increase of the fluorescent emission in the examples of S10–P33 and S25–P31. This nanoparticle size dependent bacterial membrane disruption of NPPBs is directly linked to their antimicrobial activities. As we discovered earlier in the MIC and MBC assays (Fig. 2 and Table 2), the smaller NPPBs ($d_{silica} \leq 50$ nm) capable of disrupting bacteria-mimicking GUVs (Fig. 3b) and bacterial membranes (Fig. 3c) are the ones that show strong antimicrobial activities, whereas the larger NPPBs ($d_{silica} > 50$ nm) unable to disrupt either membrane are the ones that show weak antimicrobial activities. Clearly, the hydrophilic NPPBs are membrane-active antimicrobials that selectively kill bacteria by disrupting their membranes with size dependent activities while sparring mammalian cells.

The nanoparticle-size dependent activities of NPPBs on disrupting bacterial membranes are also directly exhibited under SEM and cross-sectional TEM. Examples of SEM (Fig. 3d–g) and cross-sectional TEM pictures (h–k) of the *E. coli* control (d, h) and that incubated with S25 (e, i), S25–P31 (f, j), and S270–P32 (g, k), respectively, are shown. We reported previously that the P28 control by itself shows no bactericidal activity against both Gram− *E. coli* and Gram+ *S. aureus*, and no bacterial membrane disruption was observed under SEM or TEM when P28 was introduced to either family of bacteria[22]. Likewise, when *E. coli* is incubated with the antimicrobially inactive silica nanospheres (e.g., S25), no morphological change is observed under SEM (Fig. 3e; S25 indicated by red arrow) as compared to the *E. coli* control (Fig. 3d), and no membrane disruption is observed under cross-sectional TEM either (Fig. 3i) just like the *E. coli* control (Fig. 3h; the bacterial membrane was stained with OsO$_4$ and appears as a continuous dark layer indicated by blue arrow). The S25 that appears to adhere to *E. coli* membrane under SEM (Fig. 3e) stays with the membrane under cross-sectional TEM (Fig. 3i) with no sign of membrane disruption.

Formation of NPPBs by grafting the hydrophilic linear-chain brush polymers onto silica nanospheres doesn't alter their amicability with mammalian cells, but a fundamental transition occurs that transforms the hydrophilic "hairy" balls into potent antibiotics when the nanoparticles are small (i.e., $d_{silica} \leq 50$ nm) (Fig. 2). This nanoparticle-size dependent transformation reveals itself clearly under electron microscopes: representative SEM shows that *E. coli* cells are crumbled with S25–P31 adhering to their surfaces (Fig. 3f, nanoparticles indicated by red arrow), and cross-sectional TEM shows that the bacterial membrane is completely obliviated (Fig. 3j) with some S25–P31 (indicated by red arrow) encroaching the used-to-be cell membrane and

entering the cytosol of the bacteria, which is in sharp contrast to the S25 control that remains on the intact membrane (Fig. 3e, i). When *E. coli* cells encounter large NPPBs such as S270–P32, no bacterial morphological change is observed under SEM (Fig. 3g), and intact bacterial membrane (indicated by blue arrow) is shown under cross-sectional TEM (Fig. 3k) even when the large cationic NPPBs (indicated by red arrow) are bound to the anionic bacterial membrane (Fig. 3g, k) just like the small S25–P31 (Fig. 3f, j).

This nanoparticle-size dependent transformation of NPPBs into membrane-active antibiotics also shows up nicely for Gram+ *S. aureus*, which differs from the Gram− *E. coli* in that it has a thick nanoporous peptidoglycan encapsulation layer, and NPPBs need penetrate this capsule first in order to gain access to the bacterial membrane. Representative SEM (Fig. 3l–n) and cross-sectional TEM pictures (Fig. 3o–q) of *S. aureus* control (l, o) and that interacting with S25–P31 (m, p) and S270–P32 (n, q), respectively, are shown. Like *E. coli* (Fig. 3f, g), *S. aureus* cells are bound to the oppositely charged NPPBs due to the attractive charge interactions (Fig. 3m, n, NPPBs indicated by red arrow). Although it is difficult to tell the wellbeing of the bacteria under SEM (Fig. 3m) when compared to the *S. aureus* control (Fig. 3l), cross-sectional TEM clearly shows that while the *S. aureus* control (Fig. 3o) exhibits intact membrane (indicated by blue arrow) underneath the peptidoglycan encapsulation (indicated by black arrow), this bacterial membrane is mostly destroyed in the presence of small NPPBs such as S25–P31 (Fig. 3p). Some of the small NPPBs that cross the nanoporous peptidoglycan capsule to demolish the bacterial membrane are observed (indicated by red arrow). In contrast, when *S. aureus* encounter large NPPBs like S270–P32 (Fig. 3n, q), although the hydrophilic nanoparticles (indicated by red arrow) appear to land on the oppositely charged bacteria under SEM (Fig. 3n), they are too large to cross the nanoporous peptidoglycan encapsulation layer (indicated by black arrow) as revealed by cross-sectional TEM (Fig. 3q), and the structural integrity of bacterial membrane (indicated by blue arrow) is not compromised.

The fluorescent dye release and membrane permeation assays together with the SEM and cross-sectional TEM studies unambiguously show that while NPPBs with small nanoparticle sizes ($d_{silica} \leq 50$ nm) kill both Gram+ and Gram− bacteria by disrupting their membranes, NPPBs with larger nanoparticle sizes ($d_{silica} > 50$ nm) are either precluded from gaining access to the membrane of Gram+ bacteria, or simply unable to cause membrane disruptions even when adhered to the outer membrane of Gram− bacteria. The reason that we are still able to observe MBCs for the larger NPPBs (Table 2), albeit at high nanoparticle concentrations, is not due to bacterial membrane disruptions. Rather, it is likely caused by the densely packed NPPBs bound on bacteria (Fig. 3g, k, n, q) that impede the normal homeostasis processes critical for bacterial survival. The membrane-disruption mode of bacteria killing exhibited by small NPPBs is a powerful antibiotic action because it evades the bacterial resistance mechanisms[58,59]. As such, the small NPPBs also show potent activities against the clinical multidrug resistant *PA14* and *MU50* strains (Table 2).

**Mechanistic insight on hydrophilic NPPB nanoparticles that selectively disrupt bacterial membranes while sparring mammalian cells: the role of lipid intrinsic curvature and nanoparticle size.** Since NPPBs are cationic and the membrane potentials of live cells are negative inside, one may argue that bacterial membrane disruption is caused by electroporation, which was proposed as one possible mode of action for cationic antibacterial peptides[60]. Although electroporation can't explain

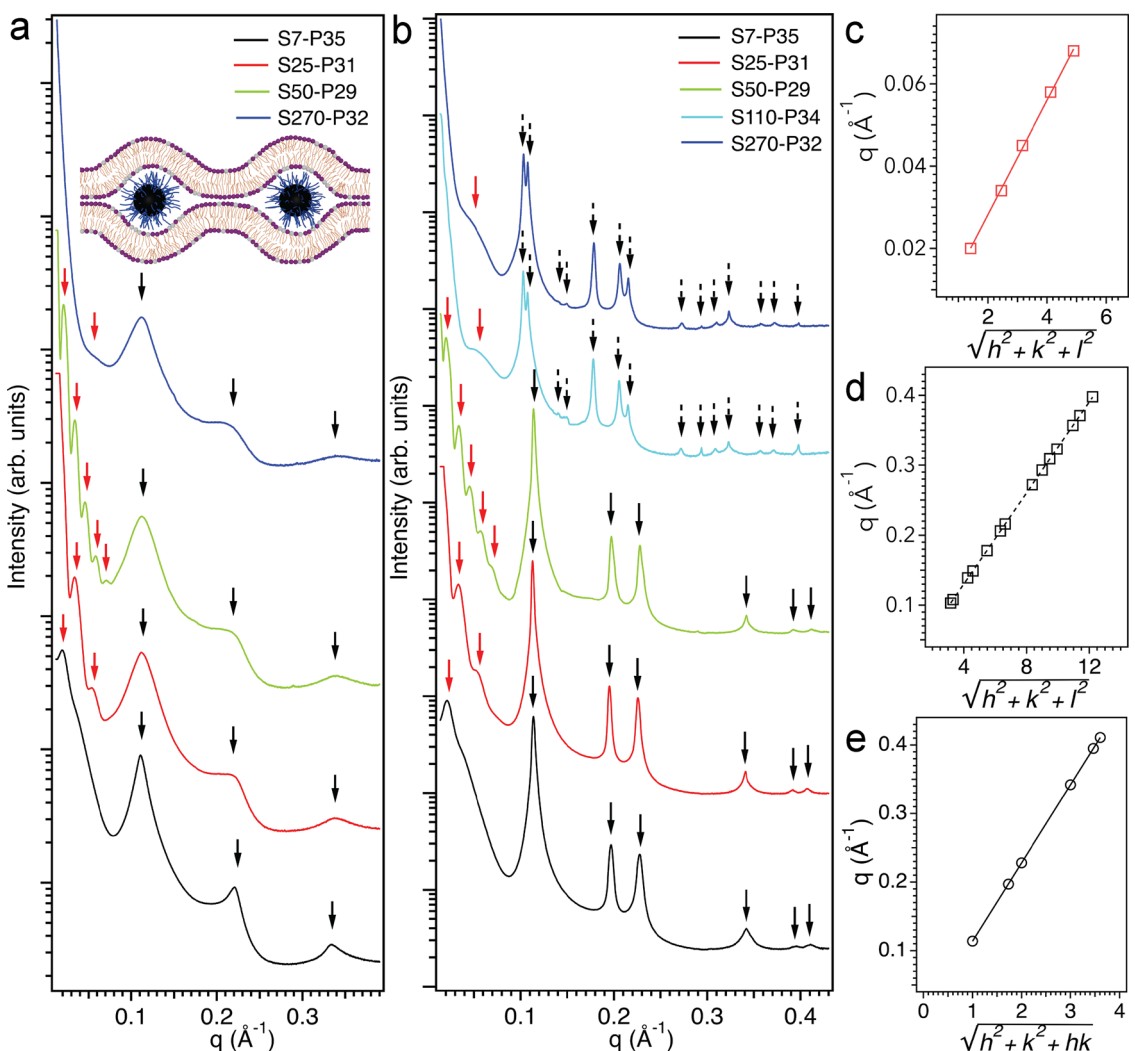

**Fig. 4 Synchrotron SAXS reveals that hydrophilic NPPBs remodel biomembranes with different modes of actions depending on both the membrane lipid intrinsic curvature and nanoparticle size. a** Mammalian cell-mimicking membrane comprised predominantly of lipids with zero intrinsic curvature are brought to stack onto each other (lamellar harmonics marked by black arrows) with the help of NPPBs adhered in the membrane "bubbles" (inset illustration: NPPBs represented by hairy balls consisting of a silica nanosphere (black) covalently grafted with P4MVP brushes (blue), and membranes represented by a binary mixture of lipids (hydrocarbon tails in golden) with both PC (purple) and PG (gray) headgroups; illustration not drawn to scale). The well-defined NPPBs themselves are ordered in 3D into cubic structures, and the cubic scatterings (marked by red arrows) show up at the same positions for the same NPPBs interacting with either mammalian cell-mimicking (**a**) or bacteria-mimicking membrane (**b**). **c** An example of the cubic scatterings from 3D ordered S50-P29 (green trace in **a**, **b**) fit to their Miller indices is shown. **b** In contrast, bacteria-mimicking membrane comprised predominantly of lipids with negative intrinsic curvature undergoes a topological transition from planar bilayer to either a 2D hexagonal ($d_{silica} \leq 50$ nm) or a 3D cubic membrane structure ($d_{silica} > 50$ nm) when encountering NPPBs of different sizes. The characteristic scatterings from the remodeled 2D hexagonal and 3D cubic membrane structures are marked by solid and dashed black arrows (**b**), respectively, and fit nicely to their Miller indices (**e**, **d**).

the observation that NPPBs selectively disrupt bacterial membrane while sparing mammalian cells, we tested this hypothesis nevertheless using carbonyl cyanide *m*-chlorophenyl hydrazone (CCCP) to dissipate the potential and pH gradient across bacterial membrane before incubating the bacteria with NPPBs. Our SEM and cross-sectional TEM studies (Supplementary Fig. 12) clearly show that targeting and disruption of both *E. coli* and *S. aureus* membranes by small NPPBs are independent on bacterial membrane potential.

Despite the simplicity of model liposomes that lack some of the conspicuous structural features of bacterial and mammalian cells, such as the membrane asymmetry, the cytoskeleton network in mammalian cells that plays important roles for endocytosis, and the cell wall structures such as the lipopolysaccharides and lipoteichoic acids of Gram− and Gram+ bacteria, respectively,

the dye release assays on model GUVs that mimic bacterial and mammalian cells still faithfully capture the fundamental membrane remodeling response when both types of cells encounter the hydrophilic NPPB nanoparticles: while mammalian membrane integrity is uncompromised regardless of the NPPB sizes, bacterial membrane can be ruptured and disintegrated when the nanoparticles are small (i.e., $d_{silica} \leq 50$ nm). The different modes of actions are corroborated by various biological assays and microscopy studies on bacterial and mammalian cells (Figs. 2 and 3), suggesting that the intrinsic curvature of membrane lipids and nanoparticle size are the two important players that synergistically define how NPPBs remodel cell membranes. We probed the membrane remodeling process at the nanoscale by synchrotron SAXS. For mammalian cell-mimicking membrane consisting predominantly of lipids with zero intrinsic curvature,

**Table 3 SAXS reveals ordered structures in bacteria-mimicking membrane remodeled by NPPBs.**

| NPPBs | Lattice ($a$, Å) | Observed peak (Å$^{-1}$)/(Miller index)/Expected peak (Å$^{-1}$) |
|---|---|---|
| S7-P35 | $H_{II}$ (63.6) | 0.114/(1,0)/0.114; 0.197/(1,1)/0.198; 0.228/(2,0)/0.228; 0.342/(3,0)/0.342; 0.395/(2,2)/0.395; 0.411/(3,1)/0.411 |
| | *ND*$^*$ | 0.021 |
| S25-P31 | $H_{II}$ (64.3) | 0.113/(1,0)/0.113; 0.195/(1,1)/0.195; 0.226/(2,0)/0.226; 0.340/(3,0)/0.339; 0.391/(2,2)/0.391; 0.407/(3,1)/0.407 |
| | Cubic (329.6) | 0.033/(1,1,1)/0.033; 0.054/(2,2,0)/0.054 |
| S50-P29 | $H_{II}$ (63.6) | 0.114/(1,0)/0.114; 0.197/(1,1)/0.198; 0.228/(2,0)/0.228; 0.342/(3,0)/0.342; 0.395/(2,2)/0.395; 0.411/(3,1)/0.411 |
| | Cubic (e.g., *Pm3n*) (450) | 0.020/(1,1,0)/0.020; 0.034/(2,1,1)/0.034; 0.045/(3,1,0)/0.044; 0.058/(4,1,0)/0.058; 0.068/(4,2,2)/0.068 |
| S110-P34 | Cubic (e.g., *Pn3m*) (193) | 0.103/(3,1,0)/0.103; 0.108/(3,1,1)/0.108; 0.139/(3,3,0)/0.138; 0.149/(4,2,1)/0.149; 0.178/(5,2,1)/0.178; 0.206/(6,2,0)/0.206; 0.216/(6,2,2)/0.216; 0.272/(6,5,3)/0.272; 0.293/(7,4,4)/0.293; 0.309/(8,5,1)/0.309; 0.323/(9,3,3)/0.324; 0.357/(10,4,2)/0.357; 0.371/(11,3,0)/0.371; 0.398/(9,8,2)/0.397 |
| | *ND* | 0.053 |
| S270-P32 | Cubic (e.g., *Pn3m*) (193) | 0.103/(3,1,0)/0.103; 0.108/(3,1,1)/0.108; 0.139/(3,3,0)/0.138; 0.149/(4,2,1)/0.149; 0.178/(5,2,1)/0.178; 0.206/(6,2,0)/0.206; 0.216/(6,2,2)/0.216; 0.272/(6,5,3)/0.272; 0.293/(7,4,4)/0.293; 0.309/(8,5,1)/0.309; 0.323/(9,3,3)/0.324; 0.357/(10,4,2)/0.357; 0.372/(11,3,0)/0.371; 0.398/(9,8,2)/0.397 |
| | *ND* | 0.043 |

Structural features attributed to remodeled membrane and NPPBs are shown in black and blue, respectively.
*ND not defined.

we observed two sets of scatterings (Fig. 4a), one of which (marked by black arrows) is equally spaced at 0.112, 0.223, and 0.335 Å$^{-1}$, respectively for all NPPBs, corresponding to a multilamellar structure with a lamellar periodicity of 56 Å, which fits nicely to a DOPG/DOPC bilayer thickness (i.e., ~44 Å[61,62]) plus a hydration layer (i.e., ~12 Å), and is attributed to stacked membranes brought together by the oppositely charged NPPBs adhered in-between the membranes. Given that we know precisely the P4MVP brush sizes, graft density, the nanoparticle sizes (Table 1), and the fact that the charge state of P4MVP is independent on pH[39], the charge density of NPPBs is calculated (i.e., ~10–30/nm$^2$ depending on the P4MVP graft density and NPPB sizes). Because the charge density of NPPBs is much higher than that of the membrane (i.e., ~1.4/nm$^2$)[62], we don't expect that NPPBs would form a continuous layer on the membrane because of the charge density mismatching[63]. In fact, the lamellar spacing does not expand when the diameter of NPPBs increases. The diffusive nature of the lamellar harmonics also indicates membrane corrugations. Taken together, we believe that the nanoparticles are wrapped in the buckled membrane "bubbles" as shown in the schematic illustration (Fig. 4a, inset). This inclusion state of NPPBs within the membrane pockets increases the entropy gain through counterion release without incurring high energy cost to break and restructure the membrane despite of their mismatched charge density. In reality though, mammalian cells would proceed to engulf NPPBs via endocytosis[33]. Nonetheless, the SAXS data still provide evidence that NPPBs do not rupture mammalian membranes regardless of their sizes.

The other set of scatterings (marked by red arrows) from the NPPB-membrane complexes varies as the NPPB size changes.

Those scatterings are attributed to the correlations from the well-defined NPPBs because the same set of correlations show up for the same NPPBs interacting with either mammalian cell-mimicking (Fig. 4a) or bacteria-mimicking membrane (Fig. 4b). For the larger NPPBs (i.e., S110–P34 and S270–P32), only one barely discernible weak nanoparticle scattering feature shows up, likely because most of the scatterings from the larger nanoparticles are beyond the range of the SAXS. For the smallest NPPB (i.e., S7–P35), only one well separated nanoparticle peak at 0.021 Å$^{-1}$ shows up too, although it is possible that additional peaks at higher $q$ are shadowed beneath the more prominent scatterings of the remodeled membranes (peaks marked by black arrows). For other small NPPBs (i.e., S25–P31 and S50–P29), multiple sharp nanoparticle diffraction peaks appear, which fit nicely to cubic structures indicative of the 3D assembly of NPPBs. For instance, the scatterings from S25–P31 are positioned at 0.033 and 0.054 Å$^{-1}$, respectively, which are related to each other by the ratio of $\sqrt{3}:\sqrt{8}$ and can be indexed as (1,1,1) and (2,2,0) of a 3D cubic structure with a lattice parameter $a$ of 329.6 Å. This unit cell matches well to the size of S25–P31 ($d_{silica}$ ~ 25 nm, Table 1) with additional spacing (~8 nm) to accommodate the bilayer and polymer brushes. Similarly, the diffractions from S50–P29 are positioned at 0.020, 0.034, 0.045, 0.058, and 0.068 Å$^{-1}$, respectively, which are related to each other by the ratio of $\sqrt{2}:\sqrt{6}:\sqrt{10}:\sqrt{17}:\sqrt{24}$ and fit nicely to the Miller indexes (1,1,0), (2,1,1), (3,1,0), (4,1,0), and (4,2,2), respectively (Fig. 4c), of the cubic structure (e.g., *Pm3n*) with a lattice parameter of 450 Å. The enlarged unit cell reflects the increased nanoparticle size of S50–P29 (i.e., $d_{silica}$ ~ 45 nm, Table 1). Although it appears to be short

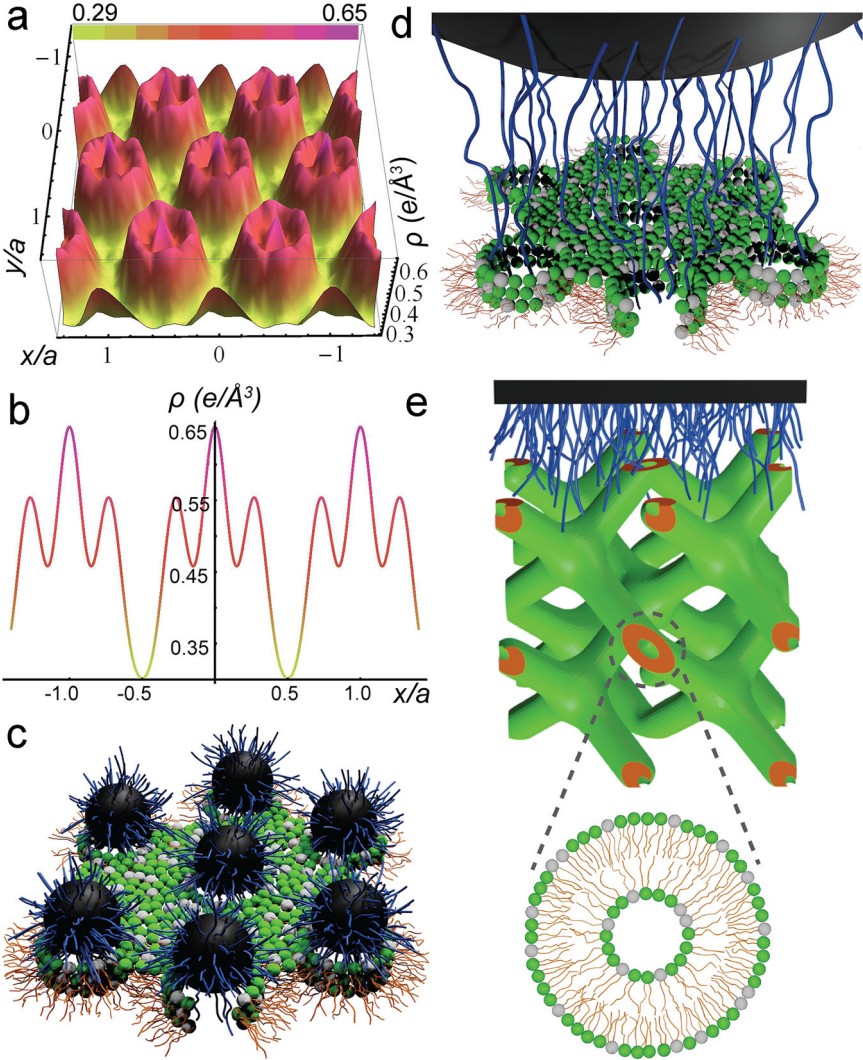

**Fig. 5 The ability of hydrophilic NPPBs to induce pore formation on bacteria-mimicking membrane is critically dependent on the nanoparticle size. a** Fourier reconstructed electron density map ($\rho$) of bacteria-mimicking membrane remodeled by S7–P35 along the lattice plane defined by *x*-axis and *y*-axis reveals that the remodeled membrane is in a honeycomb-like 2D inverted hexagonal phase ($H_{II}$), the hallmark of membrane pore formation. The electron density map is color-coded, with a scale bar on the top indicating the variation from low (yellow) to high (magenta) electron density that ranges from 0.29 to 0.65 e/Å³. **b** Quantitative analysis of the electron density distribution along the unit cell axis (e.g., *x*-axis) helps elucidate the cooperative molecular rearrangement leading to $H_{II}$ formation: attractive interactions between polymer brushes and membrane surface induce a topological transition of the membrane to form pores, where the headgroups of lipids tilt inward to encircle strands of polymer brushes ($\rho = 0.65$ e/Å³) and constitute the edge of the pores ($\rho = 0.55$ e/Å³), with their hydrocarbon tails packed in-between the pores forming the wall ($\rho = 0.29$ e/Å³). **c, d** Schematic illustrations of the $H_{II}$ formation in the presence of two different NPPBs with increasing sizes, in which the bacteria-mimicking membrane is represented by a binary mixture of lipids (hydrocarbon tails in golden) with both PE (green) and PG (gray) headgroups, and NPPBs are shown by silica nanospheres (black) covalently grafted with polymer brushes (blue). **e** When the size of NPPBs increases beyond a threshold ($d_{silica}$ ~ 50 nm), the 2D lipid $H_{II}$ phase gives way to a 3D lipid cubic phase (e.g., *Pn3m*), a unit cell of which is schematically illustrated. Inset: a cross-sectional view of the lipid bilayer tube.

of spacing for the bilayer and polymer brushes, this discrepancy could be attributed to the difference in measuring the nanoparticle size by TEM and SAXS.

For bacteria-mimicking membrane consisting predominantly of lipids with negative intrinsic curvature, besides the same set of scatterings coming from the 3D organized NPPBs that varies as the nanoparticle size changes (peaks marked by red arrows), two distinctly different sets of membrane scatterings (marked by solid and dashed black arrows, respectively) are observed (Fig. 4b). Within each set of the membrane scatterings, the SAXS pattern remain identical for different NPPBs, but there is a clear transition from one type of membrane remodeling to the other type when the NPPBs increase beyond a threshold size (i.e., $d_{silica}$

~ 50 nm). A summary of all SAXS peaks together with their Miller indices are listed in Table 3.

For small NPPBs ($d_{silica} \leq 50$ nm), a set of six membrane scatterings (marked by solid black arrows) appear, which fit nicely to a 2D hexagonal structure with a lattice size of ~64 Å (Fig. 4e). With the exception of $q_{12}$ that is too weak to be clearly identified, those scatterings represent the first seven reflections from $q_{10}$ to $q_{31}$ of the 2D membrane lattice. For large NPPBs ($d_{silica} > 50$ nm), a set of fourteen membrane scatterings (marked by dashed black arrows) appear instead, which fit nicely to a cubic structure (e.g., *Pn3m*) with a lattice parameter of 193 Å (Fig. 4d). Interestingly, the two completely different membrane remodeling behaviors are parted by a threshold nanoparticle size (i.e., $d_{silica}$ ~

50 nm) that happens to delineate NPPBs of strong ($d_{silica} \leq 50$ nm) and weak ($d_{silica} > 50$ nm) antimicrobial activities. It becomes clear that the ability to remodel the bacteria-mimicking membrane into 2D hexagonal rather than 3D cubic structures by the hydrophilic NPPBs is the harbinger of bacteria death, and this ability is critically dependent on the nanoparticle size.

To further understand the underlying mechanism of bacteria death caused by small NPPBs, we performed Fourier reconstruction of the 2D hexagonally ordered membrane using the method reported before[22,55,64]. Based on the phase criteria developed by Turner and Gruner[64], our phase choices are (+−−++−). The reconstructed electron density map of bacteria-mimicking membrane remodeled by S7–P35 along the lattice plane reveals the 2D hexagonally packed membrane pores (Fig. 5a). Since the same 2D hexagonal lattice of remodeled membrane (i.e., $a = 6.4$ nm) is observed for NPPBs of different sizes ranging from S7–P35 to S50–P29 that are all larger than the lattice size, the nanoparticles themselves should not fit into the 2D lattice. Quantitative analysis of the electron density distribution (Fig. 5b) along the unit cell axis (e.g., $x$-axis) helps elucidate the cooperative molecular rearrangement leading to the membrane pore formation. The region in-between the pores has the lowest electron density, which can be attributed to the lipid hydrocarbon tails ($\rho = 0.29$ e/Å$^3$) because they have the lowest electron density in the membrane. Approaching the rim of those pores the electron density rises to 0.55 e/Å$^3$. It then drops to 0.46 e/Å$^3$ inside the pores before a feature with the highest electron density ($\rho = 0.65$ e/Å$^3$) appears at the pore center. This electron density is much higher than that of any component in the membrane and can be only assigned to the polymer brushes because the quaternized P4MVP with associated iodide counterions has the highest electron density in the system. Notably, the high electron density confirms that even though polymer brushes are present in the pores, NPPBs are not part of the 2D lattice because the low electron density of silica nanospheres (i.e., 0.19 e/Å$^3$) doesn't show up at all. The second highest electron density ($\rho = 0.55$ e/Å$^3$) at the rim of the pores can be subsequently assigned to phospholipid headgroups because they have the next highest electron density after the polymer brushes. This electron density is higher than that of a typical phospholipid headgroup (0.41 e/Å$^3$)[64], suggesting the presence of residue iodide ions, likely due to the unmatched charge density between the polymer brushes and membrane. The presence of residue iodide ions is further confirmed by the increased electron density inside the aqueous pores ($\rho = 0.46$ e/Å$^3$), which is much higher than water by itself ($\rho = 0.33$ e/Å$^3$)[22,64]. Taken together, the Fourier reconstruction reveals a cooperative membrane remodeling pathway that starts by the attractive electrostatic interactions between polymer brushes and membrane surface, which induce a topological transition of the membrane by tilting the headgroups of surrounding lipids inward to encircle the polymer brushes in the center, and ends up with the formation of a honeycomb-like 2D inverted hexagonal phase ($H_{II}$), the hallmark of membrane pores. This mode of pore formation differs from any current mode of actions proposed for amphiphilic membrane-active antimicrobials, in which the hydrophobic moieties of those antimicrobials are expected to disrupt cell membranes by breaching into the hydrophobic membrane interior[65–67]. It is also slightly different from what we observed previously with hydrophilic bottlebrush polymers, which induce pore formation by bending the membrane to encircle themselves[22,23]. This difference is likely due to the increased sizes of NPPBs that are too large to fit into the membrane, as the polymer brushes are grown on silica nanospheres ($d_{silica} \sim 7$–270 nm) instead of the molecular backbones of bottlebrush polymers. As a subtle

additional evidence, the lattice parameter of the $H_{II}$ phase induced by the polymer brushes from NPPBs (i.e., $a = 6.4$ nm) is smaller than that by the bottlebrush polymers (i.e., $a = 7.0$ nm)[22,23]. Given that only a portion of polymer brushes on NPPBs are involved in the formation of each membrane pore, individual NPPBs may induce the formation of ca. one to several pores on the membrane depending on the nanoparticle sizes, as schematically illustrated in Fig. 5c, d, respectively. However, there is a threshold size of NPPBs beyond which it is not energetically favorable for their polymer brushes to remodel the bacteria-mimicking membrane into the $H_{II}$ phase anymore.

This threshold appears to be roughly 50 nm for the silica nanospheres. Large NPPBs ($d_{silica} > 50$ nm) remodel the bacteria-mimicking membrane into cubic structures similar to that observed when the membrane was remodeled by the linear-chain P28 control[22]. A unit cell of the bicontinuous double diamond $Pn3m$ structure is schematically shown in Fig. 5e, where interactions between the bacteria-mimicking membrane and a blanket of oppositely charged polymer brushes on a quasi-flat surface of NPPBs induce the formation of 3D bicontinuous lipid cubic phase. For illustration the bilayer tubes in the unit cell are cut open at the boundary, and a cross-sectional view of a bilayer tube is shown (inset, Fig. 5e). Unlike the 2D hexagonal $H_{II}$ phase that signifies membrane pore formation, continuous membrane still persists in the bicontinuous cubic phase. In reality though, bacterial membranes differ from the bacteria-mimicking model membrane in that the bacterial membranes are supported and constrained by bacterial cell walls. Although the polymer brush-induced formation of 2D $H_{II}$ phase and pores on the model membrane is relevant for bacterial membranes, the 3D cubic structures are less relevant because bacterial membranes are unlikely able to organize freely into 3D structures. Nevertheless, the membrane integrity difference revealed in the phase behavior of model membrane incubated with hydrophilic brush polymer controls[22,23], bottlebrush polymers[22,23], and NPPBs still reflects faithfully the fate of bacterial membranes under the same encountering events: while the formation of 2D hexagonally packed pores in the model membrane bodes well for the observed disruptions of bacterial membranes, formation of 3D cubic phases is correlated with the observed intact bacterial membranes. This observation is again different from amphiphilic membrane-active antimicrobials, where the formation of bicontinuous cubic phases correlates with membrane pore formation[67].

It's worthwhile to deliberate over the mechanism underlying the observation that no membrane disruption occurs for mammalian cells interacting with hydrophilic NPPBs of all sizes, whereas a nanoparticle size dependent membrane disruption exists for bacteria. This observation indicates that while nanostructures may help transform the non-toxic and non-bactericidal hydrophilic brush polymers into membrane-active antimicrobials, the successful outcome depends on both the intrinsic curvature of membrane lipids and the nanoparticle size. Unlike mammalian membranes that are rich in zero-intrinsic-curvature lipids, microbial membranes are laden with negative-intrinsic-curvature lipids that help generate the saddle-splay curvature critically needed for membrane pore formation[67]. When encountering the hydrophilic NPPBs, mammalian membranes buckle locally to increase the contact areas with NPPBs by enclosing them in membrane "bubbles" (Fig. 4a), which increases the attractive electrostatic interactions and entropy gain through counterion release without incurring high energy cost to break the continuity of the membranes. In contrast, microbial membranes undergo a topological transition to form 2D inverted membrane pores that wrap around individual bundles of polymer brushes with much more enhanced contact areas (Fig. 5c, d). Although formation of membrane pores maximizes the attractive

electrostatic interactions with polymer brushes and entropy gain through the counterion release, it only happens for microbial membranes because their abundant negative-intrinsic-curvature lipids help offset the energy cost to bend the membranes into nanopores via the spontaneous formation of the saddle-splay curvature. This energy cost would be prohibitively high for mammalian membranes rich in zero-intrinsic-curvature lipids. As further evidences to demonstrate the importance of lipid compositional difference that helps set apart their different paths toward different cellular fates when bacteria and mammalian cells encounter hydrophilic NPPBs, we performed SAXS studies of a series of NPPB-remodeled model membranes bearing the same charge density (i.e., 20% PG lipid) but systematically varied PE/PC ratios. As shown in Supplementary Fig. 13 for the small NPPBs ($d_{silica} \leq 50$ nm), formation of the 2D $H_{II}$ phase only occurs in the bacteria-mimicking membrane with a high PE content (i.e., ~80%). The ability to induce pore formation on the model membranes by the NPPBs subsides precipitately when the PE content drops below 70%, which includes representative membranes of eukaryotic cells.

In addition to that, there is a clear nanostructure size dependent transition on whether the hydrophilic NPPBs can induce the formation of pores on bacterial membranes. We observed previously a transition from antimicrobially-inactive hydrophilic polymers to antimicrobially-active bottlebrush polymers: while the hydrophilic linear-chain polymers interact with bacteria-mimicking membrane uniformly and induce the formation of 3D bicontinuous cubic structures without disrupting the membrane, the nanostructured bottlebrush polymers consisting of the same hydrophilic polymers grafted on molecular backbones induce the formation of 2D $H_{II}$ phase and membrane pores[22,23]. Here, we replace the soft molecular backbones with hard silica nanospheres of well-defined and systematically increased diameters, and witness a very similar transition where NPPBs of large and small diameters behave like the hydrophilic linear-chain polymers and bottlebrush polymers, respectively, in remodeling the bacteria-mimicking membrane. We attribute this transition to a change from concentrated to semidilute polymer brush regimes as the curvature of silica nanosphere increases. For NPPBs with large silica nanospheres (i.e., curvature → 0), their polymer brushes resemble those on a flat surface. At high graft density, the polymer brushes are in the concentrated regime characterized by uniformly distributed concentration along the brush height[68,69], which interact with bacterial membranes uniformly like a blanket of hydrophilic linear-chain polymers and induce the formation of 3D cubic structures on the model membrane (Fig. 5e). For NPPBs with small silica nanospheres, their curved surfaces give rise to a transition from concentrated to semidilute polymer brush regimes along the brush height[70]. Within the outer layer of NPPBs, the polymer brushes are in their semidilute regime characterized by non-uniformly distributed concentration along the brush height and increasingly relaxed conformational freedom. Even though the diameters of NPPBs are too large to fit into the membranes themselves, their polymer brushes aided by increased conformational freedom are able to interact with bacterial membranes non-uniformly as individually grouped strands, which induce the formation of 2D $H_{II}$ phase and membrane pores (Fig. 5c, d). Our observation of a size-dependent transition from concentrated to semidilute polymer brush regimes on NPPBs, or from intact bacterial membranes to disrupted bacterial membranes when the NPPBs encounter bacteria, occurs at a threshold nanoparticle size of $d_{silica}$ ~50 nm. We expect that the threshold boundary may shift depending on the polymer brush size and graft density.

In summary, we aim to clarify the myth about nanoantibiotics and illuminate a path forward for their utility in the clinical

battlegrounds by dissecting the antibiotic role of benign nanostructures from belligerent chemical moieties that indiscriminately target both bacteria and mammalian cells. Together with our previous studies on the bottlebrush polymers along this line[22,23], we come to a few preliminary conclusions.

First, nanostructures by themselves are not necessarily antibiotic, i.e., reducing the size of materials does not necessarily enhance their antibiotic activities, as witnessed by the MIC assays of bare silica nanospheres of different sizes against both Gram— *E. coli* and Gram+ *S. aureus* (Supplementary Fig. 9). It is the chemical function carried or enabled by the nanostructures rather than the physical size alone that has the potential to deliver a lethal blow to bacteria.

Second, when nanostructures come into play, benevolent chemical moieties that don't ordinarily debilitate live cells may become active antimicrobials yet remain non-toxic to mammalian cells. Nanostructures are the linchpin of this transformation, as the acquired antimicrobial activity is lost when the nanostructures fall apart[23]. We identified two important players that act synergistically underlying the transformation. On one hand, nanostructures give rise to multivalent interactions and structural rigidity needed to bend biological membranes locally upon contact, which could proceed to induce membrane pore formation; on the other hand, the intrinsic curvatures of the membrane lipids need to be conciliatory for this topological transition to happen. Unlike amphiphilic antimicrobial peptides and their synthetic mimics that use hydrophobic interactions to infiltrate and disrupt bacterial and mammalian membranes alike, the membrane disruption induced by hydrophilic nanoparticles occurs exclusively on bacteria instead of mammalian cells, because only microbial membranes laden with negative-intrinsic-curvature lipids are able to offset the energy cost to bend locally into nanopores via spontaneous formation of the saddle-splay curvature. Mammalian membranes rich in zero-intrinsic-curvature lipids can't do that because the energy cost would be prohibitively high.

Third, the nanostructure-induced antimicrobial activity of hydrophilic nanoparticles is critically dependent on nanoparticle sizes. Taking hydrophilic NPPBs as examples, although large NPPBs may adhere to bacteria and impede bacterial homeostasis processes, they don't kill bacteria by poking holes and the revealed antibacterial activity is not as potent as small NPPBs. As the size of NPPBs decreases, their polymer brushes are able to interact with bacterial membrane as individually grouped strands due to their increasing conformational freedom along the brush height, which induces pore formation and transforms the hydrophilic nanoparticles into potent antibiotics. We identified a threshold size of silica nanosphere (i.e., $d_{silica}$ ~50 nm) that sets apart the strong and weak antimicrobial activities of NPPBs; this boundary may shift depending on the polymer brush size, size distribution, graft density, etc. that all affect the transition from concentrated to semidilute polymer brush regimes along the brush height.

Finally, the physical size and shape of membrane-active antimicrobials can be used to develop nanoantibiotics with high selectivity against the two different families of bacteria. Our previous studies demonstrated the role of size on defining the selectivity of nanoantibiotics: while small hydrophilic bottlebrush polymers are potent killers for both Gram+ and Gram— bacteria, long rod-like bottlebrush polymers that are still bactericidal against Gram— bacteria become inactive against Gram+ bacteria because of the selective filter effect of their nanoporous peptidoglycan capsule[22,23]. Our study on NPPBs shed new light into this picture. Although small spherical NPPBs are indeed potent killers for both Gram+ and Gram— bacteria, large NPPBs ($d_{silica} > 50$ nm) are weak antimicrobials against both families of bacteria. We attribute the different antimicrobial selectivity

between spherical NPPBs and rod-like bottlebrush polymers to their different shapes. Because spherical NPPBs have isotropic curvatures, their polymer brushes undergo a transition from antimicrobially-active to antimicrobially-inactive states when the nanoparticle size increases. In contrast, rod-like bottlebrush polymers have anisotropic curvatures along their longitudinal and transverse directions, respectively. As long as the rod diameter is small, polymer brushes on rod-like nanoantibiotics will remain in their antimicrobially-active state even when the rod length is long.

Although our benchtop biological assays revealed the biocompatibility and nanostructure-dependent antibacterial activity of model NPPBs, in vivo studies with animal models are ultimately needed to validate their efficacy for clinical applications. We are not there yet, but our understandings on the antibiotic role of nanostructures point toward many interesting new directions for antibiotics design. For example, membrane-active antimicrobials don't have to be plagued by cytotoxicity. It's possible to exploit the nanostructure-enabled multivalent interactions to turn a wide variety of non-toxic but antimicrobially inactive polymers into potent membrane-active antibiotics without deteriorating their cordiality with mammalian cells. Those nanoantibiotics would kill bacteria upon contact yet remain non-toxic when engulfed by mammalian cells. The nanostructure-induced transformation of antimicrobial activity is independent on specific chemical structures, as we demonstrated previously that both the non-bactericidal P4MVP[22] and poly(N,N,N-trimethylamino-2-ethyl methacrylate) (PTMAEMA)[23] became potent antimicrobials following the same mechanism. Considering this modular design of nanoantibiotics, we would also envisage multifunctional nanoantibiotics in which the polymer corona helps crack the bacterial membranes while the nanoparticle core carries complementary therapeutic or diagnostic functions. The added benefits of nanostructures would also make it possible to develop antibiotics with triple selectivity, i.e., selectivity between bacteria and mammalian cells, between different families of bacteria, and between in-clinical-use and after-clinical-use states of the same antibiotics. The last concept was demonstrated recently with a nanoantibiotic design that can be dismantled and deactivated by enzymes existing exclusively in natural habitats[23]. Given the rapid progress in both the bottom-up and top-down approaches for nanomaterials discovery, our findings suggest that nanoengineering may open a promising new path for the development of clinically viable candidates of nanoantibiotics.

## Methods

**Materials**. Tetraethyl orthosilicate (TEOS, 98%), L-lysine (≥98%), L-arginine (≥98%), cyclohexane (≥98%), ammonium hydroxide (28–30% NH₃ in aqueous solution), α-bromoisobutyryl bromide (BIBB, 98%), coper(I) chloride (99%), copper(II) chloride (99.999%, trace metals basis), triethylamine (TEA, ≥99%), iodomethane (99%, stabilized with copper), potassium phosphate monobasic (KH₂PO₄, ≥99%), agar, Triton X-100, Fluorescein (95%), 3-(4,5-dimethylthiazol-2-yl)−2,5-diphenyltetrazolium bromide (MTT, 98%), glutaraldehyde solution (50%), N-Phenyl-1-naphthylamine (NPN, 98%) and hydrofluoric acid (48% HF in aqueous solution) were purchased from Sigma-Aldrich (St. Louis, MO) and used as received. 4-vinylpyridine (4VP, 95%, inhibited with 100 ppm of hydroquinone) was purchased from Sigma Aldrich and purified by passing through a column packed with base aluminum oxide before use. Tris[2-(dimethyl amino) ethyl]amine (Me₆TREN, 99+%) was purchased from Alfa Aesar (Ward Hill, MA). (3-aminopropyl)trimethoxysilane (APTES, 97%) was purchased from Gelest (Morrisville, PA) and used as received. DMP-30 and LX-112 were purchased from Ladd Research Industries (Williston, VT) and used as received. All other chemicals were ACS or HPLC grade reagents purchased from certified vendors such as Sigma-Aldrich, TCI, and Alfa Aesar, used as received unless otherwise specified.

1,2-dioleoyl-sn-glycero-3-phospho-(1′-rac-glycerol) (DOPG), 1,2-dioleoyl-sn-glycero-3-phosphocholine (DOPC), 1,2-dioleoyl-sn-glycero-3-phosphoethanolamine (DOPE), and 1,2-dioleoyl-sn-glycero-3-phosphoethanolamine-N-(lissamine rhodamine B sulfonyl) (ammonium salt) (18:1 Liss Rhod PE) were purchased from Avanti Lipid (Alabaster, AL) and used as received. E. coli (ATCC 25922) and S. aureus (ATCC 25923) were purchased from American Type Culture Collection

(ATCC) (Manassas, VA) and reactivated according to the instructions. Mueller Hinton (MH) broth was purchased from Becton, Dickinson and Company (Franklin Lakes, NJ) and used as received. The clinical multidrug resistant bacteria strains PA14 and MU50 were kindly provided by Professor Kendra Rumbaugh at TTUHSC. Fresh human red blood cells (HRBCs) were purchased from Innovative Research Inc. (Novi, MI), stored at 4 °C and used within 2 weeks. The HEK-293 (ATCC CRL-1573™) cells were kindly provided by Professor Min Kang at TTUHSC. Dulbecco's modified Eagle's media (DMEM, containing 4.5 g/L glucose, L-glutamine & sodium pyruvate) were purchased from Mediatech, Inc. (Manassas, VA). Trypsin-EDTA (0.25%, phenol red) solution was purchased from Thermo Fisher Scientific (Grand Island, NY). The live and dead staining kits for mammalian cell (Calcein-M/EthD-III) and for bacteria (DMAO/EthD-III) were purchased from Promokine (Heidelberg, Germany) and used following the instructions from the manufacture.

**Biological assays**. The MIC and MBC were determined following the method suggested by Clinical and Laboratory Standards Institute (CLSI) and literatures[40,41]. Bacteria were grown in MH broth at 37 °C for 18 h, and then diluted to fresh MH broth (100×) for re-growth. Bacterial growth was monitored by optical density at $\lambda = 600$ nm (OD₆₀₀) using a Jasco V-630 UV–Vis spectrometer from Jasco Analytical Instruments (Easton, MD) until the mid-log phase (OD₆₀₀ = 0.5–0.6) was reached.

**MIC**. Bacterial cultures at their mid-log phase were diluted by fresh MH broth to $5 \times 10^5$ CFU/mL. An aliquot (10 μL) of NPPB (or its control) with serial 2-fold dilutions and 90 μL bacterial culture in the MH broth were added into each well in a preset 96-well microplate in quadruplicate. For the positive control (i.e., PC), 90 μL bacterial culture was mixed with 10 μL MH broth. For the negative control (NC), 100 μL MH broth was used. The 96-well microplates were then incubated at 37 °C for 18 h. The OD₅₆₀ of each well was measured with a microplate reader (PerkinElmer Victor X5; Waltham, MA). The percentage of uninhibited growth ($\omega$) was calculated using Eq. (1):

$$\omega = \frac{OD_{560,sample} - OD_{560,NC}}{OD_{560,PC} - OD_{560,NC}} \times 100\%. \quad (1)$$

The MIC is defined as the concentration of NPPB (or its control) that completely inhibited bacterial growth, i.e., no optical density difference was observed within experimental error when compared to the negative control. Given that NPPBs of different sizes have different specific surface areas, the MICs of NPPBs can be further normalized to P4MVP brush concentrations for individual NPPBs when the graft density of P4MVP brushes and specific areas of NPPBs are known. As shown later in Eqs. (4) and (6), respectively, both the graft density and specific surface areas were determined by TGA.

**MBC**. Bacterial cultures at their mid-log phase were washed twice with sterile PBS buffer (10 mM KH₂PO₄, 150 mM NaCl, pH 7.4) and harvested by centrifugation at $7000 \times g$ for 5 min with a Galaxy 7D Mini Centrifuge (VWR, Radnor, PA). The harvested bacteria were re-suspended and diluted by PBS buffer to $1.5 \times 10^6$ CFU/mL. A 100 μL stock solution of NPPB (or its control) with serial 2-fold dilutions and 50 μL resuspended bacteria in PBS buffer were added into each well in quadruplicate in a 96-well microplate to reach a final bacterial concentration of $\sim 5 \times 10^5$ CFU/mL in each well. The positive control is 50 μL resuspended bacteria mixed with 100 μL PBS buffer, and the negative control is 150 μL PBS buffer by itself. The plates were incubated at 37 °C for 3 h. Serial 10-fold dilutions were subsequently made with the PBS buffer. For each dilution, 20 μL of the solution was taken and plated onto MH agar plates, which were then incubated at 37 °C overnight to yield visible colonies. The bacterial survival rate was calculated by dividing the number of colonies yielded from the bacterial growth solution at a given NPPB (or its control) concentration by that from the positive control. The negative control was used to confirm the sterility of the PBS buffer. The MBC was defined as the concentration of NPPB (or its control) at which at least 3-log reduction of bacterial survival was observed. Like MICs, MBCs can be also normalized to P4MVP brush concentrations for individual NPPBs.

**Bacterial live/dead assay**. The bacteria live/dead staining kit (PromoCell GmbH, Germany) was used to directly visualize the wellbeing of bacteria incubated with NPPB (or its control) as reported before[22]. The staining kit contains 5 mM DMAO in DMSO (Ex/Em ~ 490/520 nm) and 2 mM Ethidium Homodimer-III (EthD-III) in DMSO (Ex/Em ~ 530/635 nm). Live bacteria with intact cell membranes stain fluorescent green, whereas dead bacteria with damaged cell membranes stain fluorescent red. In a typical test, the mid-log phase bacteria were harvested by centrifugation at $7000 \times g$ and washed with sterile PBS buffer twice. The bacterial cells were then resuspended in PBS buffer and incubated with NPPB (or its control) at a chosen concentration. The resuspended bacteria incubated with sterile PBS buffer without NPPB were used as controls. After incubation for 3 h, the bacteria were harvested and resuspended in the Tris buffer (pH = 7.4, 150 mM NaCl, 10 mM Tris). The staining kit was applied to the bacterial suspension following the instruction provided by the manufacture. After staining, the bacteria were washed and re-suspended in the Tris buffer. An aliquot (10 μL) of the

bacterial suspension was imaged under a Nikon laser scanning confocal microscope (Nikon Ti-E microscope with A1 confocal and STORM super-resolution modules) running NIS-Elements AR analysis software 5.21.03 for data collection (Nikon Inc., Melville, NY) using a 100× oil-immersion objective lens.

**Bacterial membrane permeation assay.** The mid-log phase bacterial cultures were washed twice with sterile PBS buffer and harvested by centrifugation at $7000 \times g$ for 5 min. The harvested bacteria were resuspended in PBS buffer and adjusted to $OD_{600} \sim 0.5$. The NPN stock solution was prepared in acetone and kept in dark. A similar procedure as reported before was used to start the assay[23]. Briefly, a bacterial suspension (400 µL) was pipetted into a 1.5 mL Eppendorf tube kept in dark. NPN was subsequently added into the bacterial suspension kept in dark (10 µM final concentration) followed by adding NPPB. Bacterial suspension mixed with PBS buffer without NPPB was used as the negative control, whereas bacterial suspension mixed with Triton (1.0%) solution was used as the positive control. The NPN fluorescence change (Ex/Em: 350/420 nm) was recorded continuously using a F-7000 FL Spectrophotometer from Hitachi (Tokyo, Japan).

**Hemolysis assay.** As reported before[22], stock solutions of NPPB (or its control) in PBS buffer were prepared to give a range of concentrations to be tested. Fresh HRBC suspension (300 µL) was washed twice with PBS buffer (12 mL) and harvested by centrifugation at $1000 \times g$, then resuspended in PBS buffer (15 mL). Aliquots of this HRBC suspension (160 µL) were mixed with NPPB (or its control) solution (40 µL) of predetermined concentration in 1.5 mL micro-centrifugation tubes. The tubes were secured in an orbital shaker, and incubated at 37 °C at 250 rpm for 60 min. We used PBS buffer (40 µL) and Triton X-100 (40 µL, 1% v/v), respectively, mixed with HRBC suspension (160 µL) as negative and positive controls. The tubes were subsequently centrifuged at $1000 \times g$ for 5 min. Supernatant (30 µL) of each sample was diluted with PBS buffer (100 µL) and put in individual wells of a 96-well microplate. The absorbance at 415 nm was measured with a microplate reader. The percentage of hemolysis ($\psi$) was calculated using Eq. (2):

$$\psi = \frac{OD_{415,sample} - OD_{415,NC}}{OD_{415,PC} - OD_{415,NC}} \times 100\%. \qquad (2)$$

$HC_{50}$ is defined as the NPPB (or its control) concentration that causes 50% hemolysis.

**MTT.** The HEK-293 cells were grown to ~70% confluence at 37 °C in a 5% $CO_2$ incubator using tissue culture dishes and DMEM medium supplemented with 5% fetal calf serum and 0.5% Pen/Strep. After detached by Trypsin and splitting, the HEK-293 cells were seeded into a 96-well microplate at a concentration of ~$5 \times 10^3$ cells/well in fresh DMEM media (100 µL/well). Seeded cells were incubated at 37 °C in a 5% $CO_2$ incubator for about 2–3 days until the confluence reach ~70%. A graded concentration series of NPPB (or its control) solutions in DMEM were prepared and added to each well in quadruplicate (10 µL/well). For the negative and positive controls, sterile PBS buffer and 3% Triton solution, respectively, were added instead into the respective wells in quadruplicate (10 µL/well). After the 96-well microplate was incubated at 37 °C overnight, the old media were replaced by fresh DMEM (100 µL/well), and MTT stock solution in DMEM (5 mg/mL) was added into each well (10 µL/well). The 96-well microplate was incubated for another 4 h before removing the medium in each well. An aliquot of sterile DMSO (100 µL) was added into each well to dissolve the formazan crystals produced by live cells. The optical density was measured at 570 nm using the microplate reader. The percentage of cell viability ($\phi$) was calculated using Eq. (3):

$$\phi = \frac{OD_{570,sample} - OD_{570,NC}}{OD_{570,PC} - OD_{570,NC}} \times 100\%. \qquad (3)$$

**Mammalian cell live/dead staining assay.** The wellbeing of HEK-293 cells incubated with NPPBs was also directly visualized under fluorescent microscope as reported before[23]. The HEK-293 cells were seeded into a 24-well plate at a concentration of ~$5 \times 10^3$ cells/well in 100 µL of DMEM medium. The cells were incubated at 37 °C in a 5% $CO_2$ incubator for about 2 days until the confluence reach 70%. A graded concentration series of NPPB (or its control) solutions in DMEM were prepared and added to the wells in duplicates (10 µL/well). In the negative and positive controls, PBS buffer and 3% Triton, respectively, were added instead (10 µL/well). After 24 h of incubation, the cell viability was assessed using the Live/Dead Cell Staining Kit (PromoKine PK-CA707-30002) following the manufacturer's instruction. In brief, adherent cells were rinsed twice with PBS and a sufficient volume (200 µL) of calcein-AM (2 µM)/EthD-III (4 µM) staining solution was added to cover the cells. The cells were stained for 45 min at room temperature, rinsed with PBS twice, and observed under a fluorescence microscopy (Zeiss Axiovert 200M Microscope). While the cell-permeant, nonfluorescent calcein-AM is retained within live cells and converted to calcein by intracellular esterase to produce an intense uniform green fluorescence (Ex/Em ~ 495/515 nm), EthD-III enters dead cells with damaged plasma membranes to produce a bright red fluorescence (Ex/Em ~ 520/635 nm) upon binding to nucleic acids.

**Materials characterizations.** The structures of NPPBs, silica nanospheres, brush polymers as well as their interactions with bacteria or model GUVs and liposomes are characterized by a range of characterization methods.

**SEM and TEM.** The NPPBs and silica nanospheres before and after interacting with bacteria were studied following our previously reported sample preparation procedures[22]. The mid-log phase bacteria were harvested by centrifugation at $7000 \times g$ and washed with sterile PBS buffer twice. The bacterial cells were then resuspended in PBS buffer and incubated with NPPB (or silica nanosphere) at its MBC for 3 h. For the NPPB (or silica nanosphere) sample that doesn't show MBC, its concentration was set as the highest concentration of the sample being studied that shows MBC. The bacterial cells incubated with sterile PBS buffer without NPPB (or silica nanosphere) were used as controls. After the incubation, bacteria suspensions were washed with PBS buffer twice, then fixed by 2.5% glutaraldehyde solution in PBS buffer for 24 h.

To prepare SEM samples, the fixed bacteria were further washed with Millipore water three times, followed by dehydration using a series of ethanol washes and dried in a freeze dryer (Labconco, Kansas City, MO). The fixed and dried bacterial cells were placed on a carbon tape, mounted onto an aluminum stud, and coated with a thin layer of gold prior to imaging under a Hitachi S-4300 SE/N high resolution field emission SEM (Hitachi High-Tech in America, Dallas, TX) running the Quartz PCI Digital Image Capture (Quartz Imaging Corporation, Vancouver, Canada) for data collection. For NPPB and silica nanosphere before interacting with bacteria, it is directly placed, mounted, and coated in a similar manner for SEM studies.

To prepare TEM samples, the fixed bacteria were washed with PBS buffer to remove excess glutaraldehyde, further fixed with 1% osmium tetraoxide ($OsO_4$) in PBS buffer for 1 h, followed by two more washes with PBS buffer to remove excess $OsO_4$. After serial dehydrations with 25, 50, 75, 90, and 100% of ethanol, the bacteria were infiltrated with a solution of LX112 resin/acetone (weight ratio 1/2) for 2 h. The LX112 resin consists of LX112, dodecenyl succinic anhydride (DDSA), and nadic methyl anhydride (NMA) at a mass ratio of 1.8/1/0.9. The accelerator 2,4,6-tris(dimethylaminomethyl) phenol (DMP-30) was added to the resin mixture (0.14% v/v) right before use. The bacteria were further infiltrated with solutions of LX112 resin/acetone = 1/1, LX112 resin/acetone = 2/1, and 100% LX112 resin for 2 h, respectively. Finally, the bacteria were embedded in 100% LX112 resin and polymerized in 65 °C for 2 days. The solidified resin block was cut into pieces of ~80 nm thickness with an ultramicrotome equipped with a diamond blade (Reichert-Jung Ultracut E). The pieces containing the bacterial sections were stained by 4% uranyl acetate and Reynolds' lead citrate and then placed on 200 mesh copper grids, followed by imaging with a Hitachi H-8100 TEM (Hitachi High-Tech in America, Dallas, TX) running the AMT Image Capture Engine V602 (AMT Imaging Direct, Woburn, MA) for data collection. For NPPB and silica nanosphere before interacting with bacteria, it is directly placed on the TEM gids for imaging.

**TGA.** The Q500 thermogravimetric analyzer (TA Instrument, New Castle, DE) was used to measure the weight percentage loss ($w$) of the ATRP initiator or P4MVP grafted on unit mass of nanoparticles. The samples were heated from room temperature to 500 °C under $N_2$ protection at a heating rate of 10 °C/min. A similar method as reported previously[39] was adapted to obtain the graft density of P4MVP brushes. Specifically, the grafting density ($\sigma$) was calculated using Eq. (4):

$$\sigma = \left(w \cdot \frac{N}{M}\right)/S, \qquad (4)$$

where $N$ is Avogadro's number, $M$ is the number-average molecular weight of the P4MVP brushes, and $S$ is the average specific surface area of the silica nanospheres, respectively, and $w$ is the weight percentage loss of P4MVP brushes calculated using Eq. (5):

$$w = w_2 - w_1, \qquad (5)$$

where $w_2$ and $w_1$ refer to the weight percentage losses of individual NPPBs and their nanoparticle control (i.e., $SiO_2@Br$), respectively. The weight percentage loss below ~100 °C is assigned mainly to the loss of adsorbed water and other solvents, whereas that between ~200–400 °C is attributed to the degradation of P4MVP. The weight losses of APTES and initiator moieties are hidden in the background due to their small contributions, but they are expected to end before 400 °C just like the P4MVP. Besides these organic moieties, condensation and cross-linking within the $SiO_2$ nanospheres, which lead to the additional loss of $H_2O$, are expected to run continuously up to 500 °C and beyond. To minimize systematic errors, it is critical to use the difference in weight percentage losses between individual NPPBs and their $SiO_2@Br$ controls in the temperature range of ~200–400 °C to calculate the true weight percentage loss of P4MVP brushes.

The average specific surface area ($S$) of silica nanospheres was calculated using Eq. (6):

$$S = 4\pi r^2 (1-w) \Big/ \left(\frac{4}{3}\pi r^3 \rho_{SiO_2}\right), \qquad (6)$$

where $w$ is the weight percentage loss of P4MVP brushes, and $r$ represent the average radius of the $SiO_2$ nanoparticles, and $\rho$ is the density of $SiO_2$ nanospheres ($\rho_{SiO2} = 1.90$ g/mL).

**SAXS**. As reported previously[22,23], small unilamellar vesicles (SUVs) of liposomes with different lipid compositions were prepared by mixing parent lipid stock solutions in chloroform or chloroform/methanol in respective ratios. The organic solvent was evaporated by a dry $N_2$ flow, and the resultant lipid films were further dried in a vacuum. Millipore water was added to hydrate the lipid films and the final liposome concentrations were adjusted to be 20 mg/mL. After incubation at 37 °C overnight, the lipid solutions were sonicated by a probe sonicator (Sonics Vibra Cell™; Newtown, CT) to clarity and extruded through an Avanti mini-extruder set containing a polycarbonate membrane (0.1 μm pore size) for 11 times to obtain uniformly sized SUVs. The extrusions were done at room temperature, which is much higher than the gel-to-liquid phase transition temperatures of any individual lipids ($T_m$ (°C): DOPG, −18; DOPE, −16; DOPC, −17).

The self-assembled complexes comprised of anionic model liposomes and cationic NPPB with stoichiometric ratios at their isoelectric points were prepared. Samples with stoichiometric ratios above and below their isoelectric points were also prepared as controls as described before[22,23]. The complexes were subsequently transferred and sealed into quartz capillaries with a 1.5 mm nominal diameter and 10 μm wall thickness (Hilgenberg GmbH, Germany), and measured either with our in-house SAXS system or at beamline 4-2 (BL4-2) of Stanford Synchrotron Radiation Lightsource (SSRL).

The in-house SAXS is a custom-designed system built by Xenocs (Amherst, MA). It is an integrated Xeuss/BioXolver system that consists of an Eiger R 1M hybrid photon counting detector from Dectris (Baden, Switzerland), a Xenocs GeniX 3D Cu ultra-low divergence x-ray source (30 W/40 μm) coupled with the FOX3D single reflection collimating optics, scatterless slits, BioCube, BioXolver, capillary flow cell, and other accessories to operate at either (GI)SAXS, USAXS, or WAXS modes. The entire flight tube including the sample stage is under vacuum, and the adjustable sample-to-detector distance is calibrated with a silver behenate standard. The x-ray collimation, control of beam size and sample position, data collection and the radial integration of the 2D SAXS into 1D intensity profiles are performed by the SAXSLab control software suite provided by Xenocs.

The BL4-2 at SSRL uses the central part of the radiation fan produced by a 20-pole, 2 T wiggler as its source, and the incident x-ray is monochromatized by a Si(111) crystal cooled by liquid $N_2$. The end-station features a pin-hole SAXS camera with one set of scatterless beam defining/collimating slits and two sets of guard slits that define a beam size of $0.3 \times 0.3$ mm$^2$ at the sample position. The evacuated flight tube offers tunable sample-to-detector distances calibrated by a silver behenate standard. A Pilatus3 X 1M single photon counting detector from Dectris with an area of $168.7 \times 179.4$ mm$^2$ and a pixel size of 172 μm was used for data collection. The beamline and sample positions are controlled by BluIce, and the azimuthal integration that converts the 2D diffraction patterns to 1D intensity profiles are done by SAXSPipe. Both BluIce and SAXSPipe are developed and maintained by the BioSAXS Group at SSRL, and their source codes are available upon request by contacting the BL4-2 staff. A typical radiation time at SSRL is 2 s, and each sample was measured 5–10 times. No radiation damage was observed for all measurements, and no significant structural variations were observed between the same NPPB-lipid sample and its controls prepared at the stoichiometry deviated from the isoelectric point. For a subset of SAXS data revealing the 2D hexagonal pattern of remodeled membranes, Fourier reconstruction of the electron density maps was performed following the method reported before[22,55,64] using Mathematica 12 (Wolfram Research, Champaign, IL).

**Dye release assay**. Model GUVs were prepared following a previous report[55]. Briefly, a chloroform solution of lipid mixture (DOPG/DOPE = 20/80 or DOPG/DOPC = 20/80 molar ratio) that contains 0.25 mol% 18:1 Liss Rhod PE was first prepared. An aliquot of this mixture (20 μL) was spread onto a roughened and cleaned Teflon slice and dried in vacuum. After pre-hydration for 15 min under a $N_2$ flow saturated with water vapor at 50 °C, 5 mL of 100 mM sucrose that contains 40 μM Fluorescein was added as the swelling solution and incubated for 2–3 days.

To start the assay in each measurement, an aliquot of GUVs (20 μL) in 100 mM sucrose was diluted into 80 μL 120 mM glucose on a glass slide under the Nikon Ti-E laser scanning confocal microscope. After the GUVs were settled by gravity, 10 μL NPPB solution (~1 mg/mL) was added at time zero. The fluorescence change of the GUVs over time was recorded at a time interval of 30 s. Lasers at 494 nm and 558 nm, respectively, were used to excite Fluorescein and Rhodamine at low laser intensity to avoid bleach of the dyes. ImageJ (NIH, Bethesda, MD) was used to integrate the fluorescence intensity of individual GUVs in each frame recorded at different times. The background of each frame was also integrated and subtracted from the fluorescence intensity of GUVs. The percentage of dye retention was calculated by taking the ratio of the background-subtracted fluorescence intensity at different times to that at time zero, and averaged over all GUVs observed in each measurement. Error bars (i.e., standard deviation) were generated by analyzing the percentage of dye retention obtained from three independent measurements.

**Other characterization methods**. The chemical structure of brush polymers was characterized by a JEOL ECS 400 MHz $^1$H NMR Spectrometer running Delta 4.3.6 for data collection (JEOL USA, Inc., Peabody, MA), and the data were analyzed by MestReNova (version 11.0.0; MestReLab Research S.L., Escondido, CA). The chemical composition variations were studied by a Bruker Tensor 37 Fourier transform infrared (FT-IR) Spectrometer running Pus 6.5 for data

collection (Bruker Scientific LLC, Billerica, MA). The zeta potential was measured by the Malvern Zetasizer NANO running Malvern Zetasizer Software 8.01.4906 for data collection (Malvern Panalytical Inc., Westborough, MA). The molecular weight and molecular weight distribution were measured by an Agilent 1260 HPLC system (Agilent Technologies, Santa Clara, CA) with integrated Wyatt detectors (Wyatt Technology Corporation, Santa Barbara, CA). The system consists of a 1260 Infinity II Bio-Inert Pump, a 1290 Infinity Thermo-statted Column Compartment, a 1260 Infinity II Diode Array Detector, a 1260 Infinity Bio-Manual Injector, a 1260 Infinity Bioinert Fraction Collector, an Agilent PLgel 5 μm MIXED-D column ($300 \times 7.5$ mm), a Wyatt Optilab T-rEX refractive index detector, and a Wyatt miniDAWN TREOS multi-angle light scattering detector. The Agilent modules were controlled by the Agilent Chemstation OpenLAB CDS Rev. C. 01.08 (210), whereas data collection and analysis were done in Wyatt ASTRA 7. The processed data for all character-izations, unless otherwise specified, were plotted in either OriginPro 2020 (OriginLab Corporation, Northampton, MA) or IgorPro 7 (version 7.08; WaveMetrics, Portland, OR).

**Statistical analysis**. All MIC, MBC, MTT, and hemolysis tests were performed in quadruplicate, and repeated twice on different days or by different students. Mean values were reported and standard deviations were used as the error bars. For dye leakage assays from mammalian cell-mimicking and bacteria-mimicking GUVs at each time point, the mean fluorescence intensity and the standard deviation were obtained from three independent measurements.

**Reporting summary**. Further information on research design is available in the Nature Research Reporting Summary linked to this article.

## Data availability
The datasets that support the findings of this study are available from the corresponding author upon reasonable request.

## Code availability
We performed Fourier reconstruction of the 2D hexagonally ordered membrane following the method reported before[22,55,64]. The mathematical algorithm is available from the corresponding author upon request.

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

## Acknowledgements

This work was supported in part by NSF through DMR-1810767 (H.L.), by the South Plains Foundation (H.L. and H.M.) and the CH Foundation (H.L. and H.M.). We thank Prof. Kendra Rumbaugh for providing the clinical multidrug resistant *PA14* and *MU50* strains, Prof. Min Kang for providing the HEK-293 cells, Prof. Lan Guan for helpful discussions and

providing CCCP, Drs. Thomas M. Weiss, Sutomu Matsui, and Ivan Rajkovic at SSRL for help with the synchrotron SAXS experiments. SSRL is supported by the DOE Office of Basic Energy Sciences under Contract No. DE-AC02-76SF00515. The SSRL Structural Molecular Biology Program is supported by the DOE Office of Biological and Environmental Research, and by NIH-NIGMS (P30GM133894). The Pilatus detector at BL4-2 was funded by NIH (S10OD021512). The contents of this publication are solely the responsibility of the authors and do not necessarily represent the official views of NIGMS or NIH.

## Author contributions

H.L., Y.J., and H.M. conceived the idea of this study and designed the project. Y.J., W.Z., and K.T. performed the materials synthesis, characterization, and biological assays. E.K. and J.B. performed the zeta potential study. All authors contributed to data analysis. The manuscript was written through contributions from all authors.

## Competing interests

The authors declare no competing interests.
