## [Peer Review File · Nature Communications]

Hydrophilic Nanoparticles That Kill Bacteria While Sparing Mammalian Cells Reveal the Antibiotic Role of NanostructuresREVIEWER COMMENTS

Reviewer #1 (Remarks to the Author):

The authors prepared hydrophilic nanoparticles through modification of chemically inert silica nanospheres with polymer brushes (NPPBs). The hydrophilic nanoparticles could kill bacteria while sparing mammalian cells. At the same time, the mechanism of killing bacteria and sparing mammalian cells were studied in detail. I recommend it could be accepted after major revision. The comments are listed as following:

- (1)The silica nanospheres was chosen in the manu and it is undegradable in vivo. How does the NPPBs metabolize in the body?
- (2)In the manu, why choose P4MVP as polymer brushes? How about the other polymers?
- (3)How about the antibacterial and biocompatibility of NPPBs in vivo?

Reviewer #2 (Remarks to the Author):

The ms reports a comprehensive experimental study using the state-of-the-art methods, demonstrating bactericidity of silica nanoparticles grafted with P4VP cationic polymers, whilst with minimal toxicity against model mammalian cells. Neither silica NPs nor the P4VP do not show animosity against mammalian cells so the latter results are not surprising. However, the antimicrobial efficacy of the constructs appears to depend on the size of the NPs underlying the polymers. In order to shed light on the possible mechanisms of the above phenomenological observations, interactions between the same NP-P4VP conjugates and giant unilamellar vesicles (GUV) consisting of DOPG/DOPE/DOPC intended to mimic PE rich or PC rich bacterial or mammalian cytoplasmic membranes, respectively. The size dependent antimicrobial effects of NPs have been widely reported in the literature, with different mechanisms put forward. The originality of this study and insight will thus be evaluated by the results and discussions to elucidate the possible mechanism for the antimicrobial efficacy of combining silica NPs and P4VP that depends on the curvature of NPs and the apparent discrimination between bacterial and mammalian membranes. Of indeed quite a few points, I would like to focus on a few main points below.

1) The references to literature are not fully up to date and as a result the context of the study is not commensurate with the state-of-the-art knowledge. This also has clear bearing on later data discussion and interpretation benchmarked against the literature references. Even without giving any specific examples (of which there are numerous), a simple Google attempt using relevant keywords would generate a substantial list of experimental and review papers on size-dependent antimicrobial effects of NPs, cytotoxicity and cellular entry by NPs, antimicrobial nanostructured surfaces, antibacterial polymers, realistic mimetics for bacterial membranes (e.g. for scattering measurements using neutrons) in which the underlying mechanisms for similar observations are discussed and reviewed. These for instance involve the composition-dependent bending elasticity of the membranes, curvature effect of nanostructures on the architecture and conformation of polymer/protein grafts on NPs, etc. Much of these would be directly relevant to the spinning of the discussion of the results in this ms which are not anticipated in the introduction or drawn upon in the discussion.

2) The morphology of the polymer-coated NPs is well characterised in the ms. However, considering the central contributions from the charge-charge interactions at the NP-polymer-membrane interface that underpin the observations and mechanism, it is surprising that the zeta potential or the surface charge density of the NP-polymer conjugates is not reported or invoked in discussions of the mechanisms.

3) The MICs are compared in terms of the mass (i.e. ug/mL) for different particles. This would be fine for simple antimicrobial agents. However, an important consideration - along with the surface potential or charge density - is the surface area. The total surface area which reflects the availability of the P4VP with bacterial and mammalian membranes may indeed be comparable. Assuming similar mass density, one probably can scale the MIC by r^3 where r is the NP radius to

compare these values. Then there could immediately be quite a simple explanation to the size dependence.

4) The second part of the ms focusing on the interactions between NPs and GUVs is comprehensive with thorough analyses and discussions. This would formulate an independent study on the NP-model membrane interactions - which is an active and vast field. However, it does not serve adequately to probe the mechanisms of the observations above. Central to the considerations of the P4VP-membrane interactions and the bacterial-mammalian membrane differentiation is the different and unique membrane structures at the Gram+ and Gram- bacteria, which the ms alludes to in a number of places. It would dictate both the electrostatic interactions and size-dependent interactions. Then an appropriate model system is not the GUVs used, as the structural integrity (against membrane fusion) and repulsion against invading NPs or other antimicrobial agents are largely determined by the presence of lipopolysaccharides and lipoteichoic acids (respectively for Gram- and Gram+ bacterial membranes) - as many recent (and indeed ancient) studies. They also represent the key differences between bacterial and mammalian membranes. Despite lengthy discussions of the results from numerous methods, they cannot provide the mechanistic insights due to the inadequate model system design.

As such, the mechanism suggested and conclusions proposed cannot be adequately substantiated by the results presented.

Reviewer #1

(1) The silica nanospheres was chosen in the manu and it is undegradable in vivo. How does the NPPBs metabolize in the body?

We thank the reviewer for the very good consideration. Like many other nanomaterials proposed for biomedical applications, the metabolization and ultimate fate of nanomaterials including silica nanoparticles have been subjected to intensive studies in their own right (see review for example, Wan B, et al, “Metabolism of Nanomaterials *in Vivo*: Blood Circulation and Organ Clearance”, *Acc. Chem. Res.*, 46 (3), 761, 2013). Although clinical data of the long-term effect of silica nanoparticles on human health are not yet available, silica is listed amongst the “generally regarded as safe” substances by the FDA (ID Code: 14808-60-7; <https://www.fda.gov/food/generally-recognized-safe-gras/gras-substances-scogs-database>). Given that the focus of this paper is to examine the nanostructure-induced transformation of antibacterial activity, we chose the passive silica nanoparticles with well-defined but systematically varied diameters as the model system. We agree with the reviewer that studies on the metabolization of NPPBs must be in order if we proceed for clinical trials. We are not there yet but relevant data will be reported should we move toward that direction.

Revision: We have added a new paragraph with four new references (Ref# 35-38) at the beginning of the RESULTS AND DISCUSSION section, which discusses our design principle and addresses the potential safety concerns of model NPPBs.

(2) In the manu, why choose P4MVP as polymer brushes? How about the other polymers?

For membrane-active antimicrobials in general, it has been well-recognized that it is the physical property (i.e., cationic charge and hydrophobicity) rather than chemical specificity that gives rise to their disruptive interactions with cell membranes (see reviews Ref# 30-32). A wide range of cationic chemical moieties carrying primary, secondary, tertiary, and quaternary amines have been tested and there is no consensus on whether one type of cationic moiety is better than the other type. Our own studies on the hydrophilic polymer molecular brushes (see Ref# 22-23), where we tested both P4MVP or PTMAEMA as polymer brushes, have also confirmed that no difference exists on the chemical specificity of the polymer brushes. As we pointed out in the last paragraph of our manuscript, we

believe that the nanostructure-induced transformation of antimicrobial activity may “*turn a wide variety of non-toxic but antimicrobially inactive polymers into potent membrane active antibiotics without deteriorating their cordiality with mammalian cells*”.

Revision: We have added a new paragraph at the beginning of the RESULTS AND DISCUSSION section, which discusses our design principle of model NPPBs carrying the P4MVP brushes (Ref# 35-38). We have also added a statement in the last paragraph of the CONCLUSIONS section to make it clear, “*The nanostructure-induced transformation of antimicrobial activity is independent on specific chemical structures, as we demonstrated previously that both the non-bactericidal P4MVP and PTMAEMA became potent antimicrobials following the same mechanism.*”

(3) *How about the antibacterial and biocompatibility of NPPBs in vivo?*

As we explained in the answer for question #1, the focus of this study is to reveal the antibacterial role of nanostructures. We have examined thoroughly the antimicrobial activity and cytotoxicity of model NPPBs using four different bacterial strains (including two clinical multidrug-resistant bacterial strains) and two different types of mammalian cells, and performed extensive mechanistic studies to identify the antibacterial role of nanostructures. Although those benchtop biological assays that validated the antibacterial activity and biocompatibility of model NPPBs bode well for *in vivo* studies, we agree with the reviewer that *in vivo* data on animals and patients are ultimately needed if we plan to transform NPPBs from benchtop to bedside. We are not there yet, but we believe the design principles learnt from this study will help optimize NPPBs toward clinical trials.

Revision: We added a statement at the beginning in the last paragraph of the CONCLUSIONS section to reflect the reviewer’s comment, “*Although our benchtop biological assays revealed the biocompatibility and nanostructure-dependent antibacterial activity of model NPPBs, in vivo studies on animal models are ultimately needed to validate their efficacy for clinical applications. We are not there yet, ...*”

Reviewer #2

(1) *The references to literature are not fully up to date and as a result the context of the study is not commensurate with the state-of-the-art knowledge. This also has clear bearing on later data discussion and interpretation benchmarked against the literature references. Even without giving any specific examples (of which there are numerous), a simple Google attempt using relevant keywords would generate a substantial list of experimental and review papers on size-dependent antimicrobial effects of NPs, cytotoxicity and cellular entry by NPs, antimicrobial nanostructured surfaces, antibacterial polymers, realistic mimetics for bacterial membranes (e.g. for scattering measurements using neutrons) in which the underlying mechanisms for*

Department of Cell Physiology and Molecular Biophysics
STOP 6551 | Lubbock, TX 79430-6551
T 806.743.2520 | F 806.743.1512

similar observations are discussed and reviewed. These for instance involve the composition-dependent bending elasticity of the membranes, curvature effect of nanostructures on the architecture and conformation of polymer/protein grafts on NPs, etc. Much of these would be directly relevant to the spinning of the discussion of the results in this ms which are not anticipated in the introduction or drawn upon in the discussion.

We thank the reviewer for the thoughtful comments. There are indeed numerous publications in the fields of nanoantibiotics, nanotoxicity, membrane models, polymer/protein brushes on nanoparticles etc. For the interest of observing the reference number policy set by Nature Publishing Group, we only chose the most relevant references that put our work in context, which by no means are able to fully cover many important contributions in each of those fields. We will try to address the reviewer's comments one-by-one with revisions and clarifications:

- i) *Even without giving any specific examples (of which there are numerous), a simple Google attempt using relevant keywords would generate a substantial list of experimental and review papers on size-dependent antimicrobial effects of NPs, cytotoxicity and cellular entry by NPs, antimicrobial nanostructured surfaces, antibacterial polymers, ...*

We agree with the reviewer that there are quite a few reports on antimicrobial nanostructures that directly or indirectly suggested size-dependent antimicrobial effects of NPs. However, a close examination of those reports would reveal that the “*size-dependent antimicrobial effects*” of many nanostructures are rather the more general size-dependent toxicity as shown by their generation of reactive oxygen species (ROS), release of heavy metal ions, or increase of the specific hydrophobic surface area that all become prevalent at the nanoscale, which help kill bacteria but also debilitate mammalian cells. In fact, we have reviewed extensively the relevant literatures on this regard in the INTRODUCTION. For example, the first 20 references in our original submission were devoted to the relevant reviews and research articles on this topic, in which we categorized those nanostructures into nanocarriers, metal or metal oxide NPs, organic-inorganic composite NPs, graphene or graphene oxide, CNTs, dendrimers, micelles, supramolecular nanostructure, polymer molecular brushes, etc. As for cytotoxicity and cellular entry by NPs, we had included 5 relevant references, i.e., Ref# 26-29 together with a comprehensive recent review Ref# 33. We believe the discussion and relevant references on cytotoxicity and cellular entry by NPs is sufficient given that this aspect is not the focus of our study.

Revision: In view of the reviewer's comments, we think we should have done better by discussing explicitly the “*size-dependent antimicrobial effects*” reported in literatures, and including references that specifically reported the “*size-dependent antimicrobial effects*”.

We have added discussion in the INTRODUCTION that explicitly review the current understanding about the “*size-dependent antimicrobial effects*” reported in literatures with 3 new references (i.e., new Ref# 5, 7, and 13).

- ii) *...realistic mimetics for bacterial membranes (e.g. for scattering measurements using neutrons) in which the underlying mechanisms for similar observations are discussed and reviewed. These for instance involve the composition-dependent bending elasticity of the membranes, curvature effect of nanostructures on the architecture and conformation of polymer/protein grafts on NPs, etc. Much of these would be directly relevant to the spinning of the discussion of the results in this ms which are not anticipated in the introduction or drawn upon in the discussion.*

We respectfully disagree with the reviewer that “*the underlying mechanisms for similar observations are discussed and reviewed*”. The observations that amphiphilic membrane-active antimicrobials induce formation of non-lamellar structures of bacteria-mimicking membranes, such as cubic and hexagonal phases, that “*involve the composition-dependent bending elasticity of the membranes*”, are indeed abundant in literatures with many reviews (for example, Ref #65-67), but not the hydrophilic membrane-active antimicrobials, let alone the hydrophilic membrane-active antibacterial nanostructures. Our observation that hydrophilic polymer brushes on nanoparticle surface reveal a critical nanoparticle size-dependency in remodeling bacteria-mimicking membranes into different non-lamellar structures – which correlate well with their different antimicrobial activities – is not reported before. This observation is thus an important discovery for the development of novel hydrophilic nanoantibiotics that kill bacteria upon contact while remaining benevolent to mammalian cells.

As for “*realistic mimetics for bacterial membranes*”, although there is no doubt that liposomes/GUVs are valuable tools to mimic bacterial (or mammalian) membranes as witnessed by numerous reports, it should be noted that no ONE simple liposome/GUV model will answer ALL questions regarding how bacteria (or mammalian cells) respond to antibiotics, simply because it is impossible to faithfully replicate the dynamic membrane composition, membrane asymmetry, and cell wall differences among different bacterial (or mammalian) cells by liposome/GUV models. We have discussed the capability and limitation of model liposomes/GUVs throughout the manuscript. In our opinion, data derived from any liposome/GUV model must be cross-checked with actual biological assays performed on live bacteria and mammalian cells, which is what we did for this study. There have been many discussions in the past regarding the capability and limitation of liposome/GUV models as cell membrane mimics for the study of antimicrobials, in particular the choice of membrane lipid compositions. In order to gain insight on how bacteria and mammalian cells respond to membrane-active antimicrobials differently due

to their membrane lipid compositional difference, lipid bilayers enriched with zwitterionic PE and PC lipids, respectively, have been widely used as valuable models to mimic bacterial and mammalian membranes, because PE and PC lipids represent the major component of the microbial and mammalian membranes, respectively (see for example, reviews Ref# 47 and 54). For a model system designed to test the role of negative-curvature vs zero-curvature lipids (i.e., PE and PC-lipids, respectively), the two major lipid components in bacterial and mammalian cells, respectively, that sets apart the different paths toward different cellular fates when they encounter NPPBs, the use of PE-rich and PC-rich liposomes at the same charge density (i.e., by keeping the PG lipid content the same, hence eliminating any other variable in the model system) is appropriate, because addition of extra components (such as lipopolysaccharides and lipoteichoic acids for bacteria-mimicking membranes, or cytoskeleton and cholesterol for the mammalian-mimicking membranes) would unnecessarily complicate the model system and elude any meaningful data interpretation by comparing apple to orange. Following this tradition as well as the pioneer studies from Gerard Wong's lab (i.e., Ref #55, 56), we chose the PE- and PC-enriched liposomes/GUVs, respectively, to shed light on the initial membrane remodeling event due primarily to their membrane lipid difference when bacteria or mammalian cells encounter NPPBs, and cross-checked the utility of this approach by comparing the data obtained from the model membranes with those from a variety of biological assays performed on live cells.

Specifically, the results from dye leakage assays using model GUVs corroborate nicely with those from the bacterial membrane permeation assays, MIC/MBIC assays, SEM/TEM studies, live/dead cell assays, as well as hemolysis and MTT assays performed on live cells. These strong lines of evidences from different perspectives convinced us that lipid bilayers enriched with zwitterionic PE and PC lipids, respectively, are indeed valuable models to mimic bacterial and mammalian membranes, because they highlight the importance of lipid compositional difference that sets apart different paths toward different cellular fates when bacteria and mammalian cells encounter NPPBs. Consequently, liposomes comprised of the same lipid bilayers were used to help shed light on the underlying mechanism of the different membrane remodeling behaviors at the molecular level by SAXS studies.

As for “*curvature effect of nanostructures on the architecture and conformation of polymer/protein grafts on NPs*”, we have included the seminar contributions from de Gennes, Milner & Cates, Wijmans & Zhulina (see Ref #68-70) in our discussions. Since the focus of this study is not about the conformation of polymer brushes, we believe those references are sufficient in the context to understanding the size-dependent membrane remodeling behavior of polymer brushes on nanoparticle surface with different curvatures.

Revisions: We have added new discussions with four new references (Ref# 47, 48, 53, 54) in the first paragraph of the “Hydrophilic NPPBs are membrane-active antimicrobials ...” section, which explains the rationale of our choice of lipid composition for the liposome/GUV models that mimic bacterial and mammalian cells, respectively. We made it clear that (1) no one liposome/GUV model system is able to inform all aspects of responses ensued from bacteria or mammalian cells interacting with exogenous substances, (2) the liposome/GUV model system we chose is used to shed light on the initial membrane remodeling event due primarily to the membrane lipid difference when bacteria or mammalian cells encounter the membrane-active NPPBs, and (3) we cross-checked the validity of this widely used approach with biological assays on live bacteria and mammalian cells. We also added new discussions in the “Mechanistic insight on hydrophilic NPPB nanoparticles...” section with two new reviews (Ref# 65, 66) to compare the antimicrobial mode of actions we discovered for the hydrophilic NPPBs with those previously proposed for the amphiphilic membrane-active antimicrobials.

(2) The morphology of the polymer-coated NPs is well characterised in the ms. However, considering the central contributions from the charge-charge interactions at the NP-polymer-membrane interface that underpin the observations and mechanism, it is surprising that the zeta potential or the surface charge density of the NP-polymer conjugates is not reported or invoked in discussions of the mechanisms.

We did report the charge density of the NPPBs, compare that with the charge density of lipid bilayers, and discuss its implications on understanding the co-assembly mechanism between NPPBs and the lipid bilayers (see discussions in the second paragraph of the “Mechanistic insight on hydrophilic NPPB nanoparticles...” section). Since we determined precisely the polymer brush size, graft density, and the nanoparticle size, we were able to calculate the charge density of NPPBs straightforwardly because each P4MVP repeating unit carries one charge.

As for the zeta-potential data, we agree with the reviewer that comparing the zeta potentials of different NPPBs would be helpful as they are qualitatively related to the sizes of the charged polymer brushes. However, quantifying surface charge density from zeta potential would be an overstretch, because zeta potential only reflects the potential at the slipping plan of the electric double layer. Not only do the measured zeta potential values depend highly on the ionic strength and buffer conditions, no reliable method exists to obtain *surface potential* from the *zeta potential*. Note it is the *surface potential* rather than the *zeta potential* that is linked to the surface charge density (for example, in the Gouy-Chapman theory that assumes a flat surface and symmetrical electrolyte). Even if we extrapolate a series of zeta potential values measured at different ionic strengths to reach a somewhat approximate value of the surface potential, the Gouy-Chapman theory would not apply

here because of the curved and “fuzzy” nature of the nanoparticle surfaces due to the presence of polymer brushes.

Revisions: We measured the zeta potentials of NPPBs and confirmed that all NPPBs have similar zeta potentials, which reflects their similarly sized P4MVP brushes. The zeta potential data is now included as the new Figure S4. We added discussions about the zeta potential data of NPPBs in the “Synthesis and characterization of hydrophilic NPPBs” section. We also revised the second paragraph of the “Mechanistic insight on hydrophilic NPPB nanoparticles...” section to explicitly discuss the charge density of NPPBs and its role on the underlying co-assembly mechanism between NPPBs and the lipid bilayers.

(3) The MICs are compared in terms of the mass (i.e. ug/mL) for different particles. This would be fine for simple antimicrobial agents. However, an important consideration - along with the surface potential or charge density - is the surface area. The total surface area which reflects the availability of the P4VP with bacterial and mammalian membranes may indeed be comparable. Assuming similar mass density, one probably can scale the MIC by r^3 where r is the NP radius to compare these values. Then there could immediately be quite a simple explanation to the size dependence.

We agree with the reviewer that the MICs of NPPBs can be represented either in mass concentration or more interestingly, the surface concentration of the P4MVP brushes on the NPPBs. This is exactly what we reported in Figure 2. Note that in the second representation of MICs, the available P4MVP brushes are normalized by the surface area of the NPPBs at the MICs. Since we determined precisely the nanoparticle size, the P4MVP brush size, molecular weight and the graft density, we were able to convert the measured MICs of NPPBs to the surface concentration of P4MVP as long as we know the specific surface areas of NPPBs. Our method to calculate the specific surface areas of NPPBs was explained in the Materials and Method section. All the MICs represented this way also revealed nicely the nanoparticle-size dependency as the two examples shown in Figure 2 d & e, which we discussed in the “Antimicrobial activity and cytotoxicity of model NPPBs” section.

Revisions: N/A.

(4) The second part of the ms focusing on the interactions between NPs and GUVs is comprehensive with thorough analyses and discussions. This would formulate an independent study on the NP-model membrane interactions - which is an active and vast field. However, it does not serve adequately to probe the mechanisms of the observations above. Central to the considerations of the P4VP-membrane interactions and the bacterial-mammalian membrane differentiation is the different

and unique membrane structures at the Gram+ and Gram- bacteria, which the ms alludes to in a number of places. It would dictate both the electrostatic interactions and size-dependent interactions. Then an appropriate model system is not the GUVs used, as the structural integrity (against membrane fusion) and repulsion against invading NPs or other antimicrobial agents are largely determined by the presence of lipopolysaccharides and lipoteichoic acids (respectively for Gram- and Gram+ bacterial membranes) – as many recent (and indeed ancient) studies. They also represent the key differences between bacterial and mammalian membranes. Despite lengthy discussions of the results from numerous methods, they cannot provide the mechanistic insights due to the inadequate model system design. As such, the mechanism suggested and conclusions proposed cannot be adequately substantiated by the results presented.

We thank the reviewer for the thoughtful comments. As we explained above in our answer to his/her comment (1)-ii), while no ONE simple liposome/GUV model would answer ALL questions regarding how bacteria (or mammalian) cells respond to antibiotics in the real-world scenarios, lipid bilayers enriched with zwitterionic PE and PC lipids, respectively, have been widely used to mimic bacterial and mammalian membranes and generated valuable insights on the membrane remodeling event due primarily to their membrane lipid difference when bacteria or mammalian cells encounter antimicrobials. We have validated this widely used approach by cross-checking the results from model GUVs with extensive biological assays performed on live bacteria and mammalian cells. Of particular importance, our synchrotron SAXS studies using model liposomes show that there is no pore formation for the mammalian-mimicking membrane encountering NPPBs of any sizes, while a critical nanoparticle-size-dependent pore formation exists for the bacteria-mimicking membrane. This observation highlights for the first time how both the nanoparticle size and membrane lipid compositional work synergistically to yield very different membrane remodeling outcomes when bacterial and mammalian cells encounter the hydrophilic nanoparticles, which in our opinion, will have far-reaching implications for the development of hydrophilic nanoantibiotics. As we discussed throughout the manuscript, despite the simplicity of the model liposomes that lack some of the conspicuous structural features of bacterial and mammalian cells, this molecular lever insight on membrane remodeling difference is completely in line with the experimental observations of live bacteria and mammalian cells that are put in contact with NPPBs in various biological assays. With that said, we were also cautious about the limitation of the model liposomes/GUVs, and discussed in length that we should not over-interpretate the data from the model membranes. For example, we specifically pointed out that while the NPPB-induced formation 2D packed membrane pores in model bacteria-mimicking liposomes have relevancy for bacteria, the 3D bicontinuous lipid phase in model membranes is less relevant because bacterial membranes are unlikely able to organize

themselves into 3D structures due to the presence of lipopolysaccharides and lipoteichoic acids in their cell walls. While we could continue debating on which liposome model would be a better choice to mimic bacteria or mammalian cells (there have been plenty of deliberations in literatures, see for example review #48), the bottom line is no liposomes or GUVs are able to faithfully replicate all structural features of live cells, and the choice of a particular liposome/GUV model would depend on the questions we aim to answer. For the purpose of testing the role of negative-curvature vs zero-curvature lipids that sets apart the different membrane disruption outcomes of bacterial and mammalian cells, we can't deny the strong correlations between the results from those widely used liposome models and the actual experimental observations on live cells.

As for the presence of lipopolysaccharides and lipoteichoic acids for Gram- and Gram+ bacterial membranes, respectively, we agree with the reviewer that they are not simple bystanders when bacteria encounter the hydrophilic nanoparticles. As we discussed in the manuscript, we proposed that the nanoporous lipoteichoic acid encapsulation layers in the Gram+ bacteria act as selective filters that prevent larger nanoparticles from gaining access to the bacterial membranes. We presented compelling evidences in Figure 3 and Figure S8 to validate this hypothesis in addition to supportive data from other biological assays. For the lipopolysaccharides in the OM of Gram- bacteria, while their interactions with NPPBs are expected since they are anionically charged just like the PG lipid, their presence doesn't change the fact that PE lipid is still the major component in the bacterial membranes.

Revisions: In light of the reviewer's comment, we performed additional SAXS studies on model membranes with systematically varied PE/PC ratios to demonstrate the importance of lipid compositional difference between bacterial and mammalian membranes that sets apart their different paths toward different cellular fates when bacteria and mammalian cells encounter NPPBs. This study is now included as the new Figure S9.

Editors

(1) We agree with the concerns over the use of the GUV as a model system.

We have made revisions and clarifications with new supportive data. Please refer to our answers for Reviewer #2, comment (1)-ii) and comment (4).

REVIEWER COMMENTS

Reviewer #1 (Remarks to the Author):

The author have replied the comments in detail. I think it could be accepted with present version.

Reviewer #2 (Remarks to the Author):

The objections from the reviewers and the rebuttal from the authors seem to focus - and remain divergent - on a few points. Without re-reviewing the ms, these points are as follows.

1) The choice of the polymer to be anchored on the nanoparticles. The authors referred to their previous studies of these polymers - in essence, their antimicrobial efficacy has already been previously demonstrated. That is, they have studied them before and so would use these polymers again here. This does not seem to address reviewer's question on the merit of the polymer coating in the broader context.

2) Setting the current study appropriately in the context by making contact with relevant literature. The authors dismissed reviewer's comments by referring to the restriction on the number of references allowed by the journal and they believe most relevant references had already been included. This does not seem to be the case. The polymers themselves and their antimicrobial effect are not new. As such, it is their conjugation with nanoparticles to render them a nanoobject that differs from author's previous studies. Interactions between such nanoparticles - nude or decorated with different polymers (cationic polymers in particular as used in the ms) - and membranes, cells, and bacteria are well reported in the literature. Beyond failing to give credit where it is due in places, the discussions and appreciation for the novelty of the work also suffered as a result.

3) The novelty on the size-dependent antimicrobial effect. The toxicity, physical properties, and interactions with biological systems of nanoparticles depend on their size - this is well known. The fact that the "biocompatible polymer" coating in this study is benign to mammalian cells is not surprising. Their antimicrobial efficacy must be related to the surface chemistry and structure of the bacterial membranes - how it differs from the mammalian cell wall, and how the polymer coating must exhibit different conformation or charge behavior depending on the size. The authors in their rebuttal made it clear that the MIC referred to the available surface concentration of the polymer brushes. As such, the size-dependent effect remains unclear - whether it is just a surface area effect or size-induced polymer conformation or interaction effect.

4) Most importantly the validity and relevance of the mechanistic study in the 2nd part of the ms to the phenomenological observations in the first part. The author went to some length to acknowledge the limitations of their model GUVs and what they might be useful for. However, the key point is that they lack the key structural components to mimic the antimicrobial membranes. This means that the interactions between nanoparticles and these GUVs do not represent those at bacterial membranes, and thus the GUV results cannot substantiate the mechanisms underpinning the observations in the first part. It is well established that nanoparticles could induce curvature in model membranes or lipid mesophases, and the results from this ms also seem to agree with these established findings. However, it serves inadequately - very much so - to yield the mechanistic insights as intended.

In summary, the ms - detailed and comprehensive in reporting its comprehensive experiments and results - utilized polymers that are known to be biocompatible and antimicrobial, coated on nanoparticles leading to effects that might be size-dependent or surface area dependent, with a separate section using model membrane systems (despite the recent advances in development of more realistic model systems as pointed out by the reviewer) that yield the results not directly

relevant to the bacterial membrane rupture. As such, it would be premature to recommend its publication before these issues are adequately addressed.

TEXAS TECH UNIVERSITY
HEALTH SCIENCES CENTER™

Reviewer #1

(1) The author have replied the comments in detail. I think it could be accepted with present version.

We thank the reviewer for his/her constructive comments that helped improve our paper. We are glad to hear that he/she thinks the paper is ready for publication at its present version.

Reviewer #2

(1) The choice of the polymer to be anchored on the nanoparticles. The authors referred to their previous studies of these polymers - in essence, their antimicrobial efficacy has already been previously demonstrated. That is, they have studied them before and so would use these polymers again here. This does not seem to address reviewer's question on the merit of the polymer coating in the broader context.

We are not aware that the merit of our model polymer brush was ever a concern in the last review. Reviewer #2 did not comment on this aspect; Reviewer #1 wondered why P4MVP was chosen as the model polymer brush, and asked whether other polymers could be used as well. We have addressed this question with clarifications that have satisfied Reviewer #1.

As explained clearly in our manuscript, the novelty of this work is NOT about reporting a new antimicrobially-active polymer. Rather, it is the discovery of the antibiotic roles of benign nanostructures that help transform the nontoxic yet antimicrobially-inactive polymers into potent antibiotics. This discovery is significant because it opens a new path toward developing clinically viable nanoantibiotics that bust bacteria upon contact while remaining nontoxic when engulfed by mammalian cells. The novel contribution of this work as well as its relations with our previous reports have been explained in details in the paper.

(2) Setting the current study appropriately in the context by making contact with relevant literature. The authors dismissed reviewer's comments by referring to the restriction on the number of references allowed by the journal and they believe most relevant references had already been included. This does not seem to be the case. The

polymers themselves and their antimicrobial effect are not new. As such, it is their conjugation with nanoparticles to render them a nanoobject that differs from author's previous studies. Interactions between such nanoparticles - nude or decorated with different polymers (cationic polymers in particular as used in the ms) - and membranes, cells, and bacteria are well reported in the literature. Beyond failing to give credit where it is due in places, the discussions and appreciation for the novelty of the work also suffered as a result.

This appears to be a moot point because we have added 15 new references in our last revision to address the reviewer's comment. The revised manuscript has reached the maximum allowable number of references for research articles set by Nature Publishing Group. We agree with the reviewer that there are many publications in the broad field of nanoparticle–membrane or cell interactions, and we have made our best effort to discuss those most relevant references that help the audience understand our particular work, that is, the membrane remodeling behaviors of hydrophilic nanoparticle-pinned polymer brushes, in contexts without breaking the reference number policy. We are cautious that we might still have missed some important works and would appreciate the reviewer to give us specific examples.

On a side note, the reviewer repeatedly referred our nanoparticle-pinned polymer brushes as “*polymer coating*” or “*decorated polymer*”. We'd like to clarify that polymer brushes are conceptually different from “*polymer coating*” or “*decorated polymer*” due to their unique conformation, which is a key determining factor for their nanoparticle-size-dependent membrane remodeling behaviors as explained in the manuscript. We'd also like to point out that although the term “*cationic*” has been used in the titles of a few antibacterial studies in which nanoparticles were modified by polymers or small molecules, a close examination of those papers would reveal that the *cationic* moieties are actually cationic and amphiphilic, where a careful balance of cationic charge and hydrophobicity was often involved. We have in fact discussed some of those studies (i.e., ref# 8-10) in our original submission.

(3) The novelty on the size-dependent antimicrobial effect. The toxicity, physical properties, and interactions with biological systems of nanoparticles depend on their size - this is well known. The fact that the "biocompatible polymer" coating in this study is benign to mammalian cells is not surprising. Their antimicrobial efficacy must be related to the surface chemistry and structure of the bacterial membranes - how it differs from the mammalian cell wall, and how the polymer coating must exhibit different conformation or charge behavior depending on the size. The authors in their rebuttal made it clear that the MIC referred to the available surface concentration of the polymer brushes. As such, the size-dependent effect remains

unclear - whether it is just a surface area effect or size-induced polymer conformation or interaction effect.

As explained in our original submission, the MICs were represented by either the concentrations of NPPBs or normalized polymer brush concentration on the NPPBs. Both representations revealed the same nanoparticle-size dependent antimicrobial activity. Note that the second representation was specifically designed to rule out the size-dependent *surface area effect* as pointed out by the reviewer, as the available polymer brush concentrations are normalized by the surface area of the NPPBs at the MICs. Regrettably, the reviewer still missed this important point.

(4) Most importantly the validity and relevance of the mechanistic study in the 2nd part of the ms to the phenomenological observations in the first part. The author went to some length to acknowledge the limitations of their model GUVs and what they might be useful for. However, the key point is that they lack the key structural components to mimic the antimicrobial membranes. This means that the interactions between nanoparticles and these GUVs do not represent those at bacterial membranes, and thus the GUV results cannot substantiate the mechanisms underpinning the observations in the first part. It is well established that nanoparticles could induce curvature in model membranes or lipid mesophases, and the results from this ms also seem to agree with these established findings. However, it serves inadequately - very much so - to yield the mechanistic insights as intended.

We respectively disagree with the reviewer's statement that "*It is well established that nanoparticles could induce curvature in model membranes or lipid mesophases, and the results from this ms also seem to agree with these established findings*". For example, we are not aware of any prior studies that demonstrated nanoparticles are able to remodel bacteria-mimicking membranes into a unique 2D columnar phase in a size-dependent manner that coincided with membrane pore formation on bacteria. We would appreciate the reviewer to point to us specific references that show our discovery "*agree with these established findings*". In addition and in contrast to the reviewer's suggestion, we'd like to point out that an important discovery of this work is that nanoparticles themselves cannot directly participate in the formation of the 2D columnar phase that we observed – they are simply too large to be able to fit into the membrane structure in any physically possible way. Rather, it is the polymer brushes on the nanoparticles that help remodel bacteria-mimicking membranes into the porous structure, and this unique membrane remodeling behavior is dependent on both nanoparticle sizes and lipid curvatures. To the best of our knowledge, no "*established findings*" of similar kind have ever been reported or known before this study.

Regarding the validity and relevance of using model liposomes as mimics for bacterial and mammalian cell membranes, we have made it clear in our revised manuscript and rebuttal letter that no model liposome capable of replicating all important structural features of cell membranes exists, and no ONE liposome/GUV model would answer ALL questions regarding how bacterial (or mammalian) cells respond to antibiotics in the real-world scenarios. Instead of repeating our explanations, we'd just like to briefly reiterate that the validity of any liposome/GUV models should depend on specific research question and cross-checked with actual biological assays performed on live cells, which is what we did for this study. Rather than dismissing those models altogether in spite of their broad utility in biophysics, biochemistry, biomaterials, and bioengineering that has helped generate invaluable insights on membrane biology, the real question is whether results derived from a liposome/GUV model corroborate with experimental observations performed on live cells. For the purpose of testing the role of negative-curvature vs zero-curvature lipids that sets apart the different membrane disruption outcomes of bacterial and mammalian cells, we have confirmed the strong correlations between the results obtained from those widely used liposome models and the actual experimental observations made on live cells. As such, we remain confident that the novel mechanistic insight garnered from our model studies is relevant and important, as it illuminates a new design concept for the development of clinically viable nanoantibiotics.

REVIEWERS' COMMENTS

Reviewer #2 (Remarks to the Author):

NA

Reviewer #3 (Remarks to the Author):

The authors present their discovery that polymer-brush conjugated nanoparticles smaller than a specific size are able to simultaneously rupture bacterial cell membranes while remaining benign to mammalian cells. The finding will clearly motivate many other groups to follow their lead in developing safe bioactive antimicrobials resulting from the physical properties of the NP and the conjugated polymer conformation rather than the more traditional biochemical approach to antimicrobial development where the chemical structure of the drug-macromolecule and its interactions with cell molecules is central to bioactivity. The key hypothesis (based on model studies with distinct GUVs) is that the physical large-scale response of the bacterial membrane upon contact with the polymer-brush NP is important for membrane disruption and antimicrobial properties.